

# On the multi-day haze in the Asian continental outflow: An important role of synoptic condition combined with regional and local sources

Jihoon Seo[1,2], Jin Young Kim[1], Daeok Youn[3], Ji Yi Lee[4], Hwajin Kim[1], Yong Bin Lim[1], Yumi Kim[5], Hyoun Cher Jin[1]

[1]Green City Technology Institute, Korea Institute of Science and Technology, Seoul, 02792, South Korea
[2]School of Earth and Environmental Sciences, Seoul National University, Seoul, 08826, South Korea
[3]Department of Earth Science Education, Chungbuk National University, Cheongju, 28644, South Korea
[4]Department of Environmental Engineering, Chosun University, Gwangju, 61452, South Korea
[5]Division of Resource and Energy Assessment, Korea Environment Institute, Sejong, 30147, South Korea

*Correspondence to*: Jin Young Kim (jykim@kist.re.kr)

**Abstract.** Air quality of the megacities in the populated and industrialized regions like East Asia is affected by both local and regional emission sources. A combined effect of regional transport and local emissions on multi-day haze was investigated by synthetic analysis of $PM_{2.5}$, sampled at both an urban site in Seoul, South Korea and an upwind background site in Deokjeok Island over the Yellow Sea, during a severe multi-day haze episode in late February 2014. Inorganic components as well as carbonaceous species of daily $PM_{2.5}$ samples were measured, and gaseous pollutants, local meteorological factors and synoptic meteorological conditions were also determined. Dominance of fine-mode particles, a large secondary inorganic fraction (76%), high OC/EC ratio (7.3), and highly oxidized aerosols under relatively warm, humid, and stagnant conditions characterize the multi-day haze episode; however, the early and late stages of the episode show different chemical compositions of $PM_{2.5}$. High concentrations of sulfate in both Seoul and the upwind background in the early stage suggest a significant regional influence on the onset of the multi-day haze. At the same time, high concentrations of nitrate and organic compounds in Seoul, which are local and highly correlated with meteorological factors, suggest the contribution of local emissions and secondary formation under the stagnant meteorological condition to the haze. A slow eastward-moving high-pressure system from southern China to the East China Sea induces the regional transport of aerosols and potential gaseous precursors for secondary aerosols from the North China Plain in the early stage but provides stagnant conditions conducive to the accumulation and the local formation of aerosols in the late stage. A blocking ridge over Alaska developed during the episode hinders the zonal propagation of synoptic-scale systems and extends the haze period to several days. This study provides chemical insights of haze development sequentially by regional transport and local sources, and shows that the synoptic condition plays an important role for the dynamical evolution of long-lasting haze in the Asian continental outflow region.

# 1 Introduction

Haze is an atmospheric phenomenon causing visibility impairment primarily resulting from scattering and absorbing light by particulate matter (PM) in ambient air such as dust, smoke, and other organic and inorganic aerosols. Haze often accompanies





the high concentrations of gas pollutants and fine-mode PM with diameter of 2.5 μm or less ($PM_{2.5}$) and has adverse effects on the human respiratory and cardiovascular systems (Pope and Dockery, 2006; WHO, 2006; EPA, 2012). In addition, low visibility and solar dimming by severe haze could pose hazards to both land and air traffic, reduce crop yields (Chameides et al., 1999), and affect both meteorology and climate by changing radiation budget (Ramanathan et al., 2001; Wang et al., 2014b).

Recently the East Asian countries, especially China, have suffered from the regional-scale prolonged haze during the cold season (Tao et al., 2012; Wang et al., 2014a; Jiang et al., 2015), thus have faced serious public health and economic risks (Gao et al., 2015). Such severe long-lasting events are driven by the combination of synoptic meteorological conditions and secondary aerosol formation processes (Zhao et al., 2013; Zheng et al., 2015). Chemical speciation and source apportionment studies have shown that secondary aerosols rather than primary aerosols mostly contribute to the fine particles during the

prolonged haze (Huang et al., 2014; Sun et al., 2014; Zhang et al., 2014). Synoptic meteorological conditions conducive to accumulation of the primary pollutants and gaseous precursors for secondary aerosols, regional transport among the megacities, and secondary aerosol formation through heterogeneous surface reactions or multiphase aqueous chemistry play critical roles in development and maintenance of the widespread and durable haze (Zhao et al., 2013; Wang et al., 2014c; Zheng et al., 2015).

The long-lasting haze with high $PM_{2.5}$ concentrations has also been a matter of concern in South Korea, the geographical neighbor of China across the Yellow Sea (Kang et al., 2004; Kang et al., 2013; Park et al., 2013; Shin et al., 2014; Park et al., 2015; Kim et al., 2016). The multi-day PM pollution in the capital city of Seoul has steadily occurred during the cold season (Oh et al., 2015) despite the decrease in PM concentration since the 2000s owing to reduction of diesel vehicles emissions and fugitive dust (Ahmed et al., 2015; Ghim et al., 2015).

The cause of the multi-day haze episodes in Seoul could be quite complex since the air quality in Seoul and its metropolitan area (Seoul Metropolitan Area; SMA) could be affected by both local emissions and transport by the Asian continental outflow. The SMA is a highly populated and industrialized region with a population of 25 million, 9 million vehicles, and 49% of gross domestic product (GDP) although it is only 12% of South Korea's land area. A large proportion of South Korea's emissions of primary pollutants and secondary aerosol contributors are emitted from the SMA: 38% of carbon monoxide (CO), 32% of

volatile organic compounds (VOCs), 26% of nitrogen oxides ($NO_x$), and 19% of ammonia ($NH_3$) (NIER, 2015). The fraction of sulfur oxide ($SO_x$) emission in the SMA is small (only 9% of the total South Korea's emission) due to the expansion of natural gas and low-sulfur fuel usage since the late 1990s (Kang et al., 2006). Besides domestic emissions, the background air pollution levels in SMA could be affected by massive emissions in the Chinese eastern coastal region. The emission intensities of the Jing-Jin-Ji (Beijing-Tianjin-Hebei), Shandong, Jiangsu, and Shanghai (a total area of 482,600 $km^2$) in 2010 were

estimated as high as 98.1 t $km^{-2}$ for CO, 17.4 t $km^{-2}$ for $NO_x$, 14.3 t $km^{-2}$ for sulfur dioxide ($SO_2$), 13.7 t $km^{-2}$ for VOCs, and 4.8 t $km^{-2}$ for $NH_3$, and these are much larger than those of South Korea (an area of 100,200 $km^2$) estimated as 8.4 t $km^{-2}$ for CO, 10.6 t $km^{-2}$ for $NO_x$, 4.2 t $km^{-2}$ for $SO_2$, 8.5 t $km^{-2}$ for VOCs, and 1.9 t $km^{-2}$ for $NH_3$ (Li et al., 2017). The prevailing midlatitude westerlies combined with the massive emissions in China often result in the transboundary transport of regional pollutants to the Korean Peninsula (Kim et al., 2007; Lee et al., 2013; Kim et al., 2014). A recent source apportionment study



on $PM_{2.5}$ in Seoul has estimated that, in average, about 30% of $PM_{2.5}$ mass is attributed to regional sources, excluding potential contributions from transported secondary aerosol precursors (Kim et al., 2016).

In such a complex source region, synoptic meteorological conditions could be an important factor for the long-lasting haze pollution. A composite analysis for the multi-day high $PM_{10}$ episodes in Seoul shows that a strong high-pressure system over
the eastern China–Korean peninsula region, which is slowly developed and moved from the central China, could trap the pollutants over China within the boundary layer and gradually spread them into the downwind region by weak westerlies (Oh et al., 2015). Although Oh et al. focused more on the regional effects caused by such synoptic pattern, the stagnant high-pressure system could also provide a conducive environment for accumulation of primary pollutants and secondary aerosol precursors. Subsidence in the high-pressure region strengthens an inversion layer, suppresses boundary layer growth, and
reduces boundary layer height (Stull, 1988; Angevine et al., 1994). In addition, local meteorological factors like insolation, temperature, and relative humidity (RH), which affect the chemical processes for secondary aerosol formation and particle aging (Seinfeld and Pandis, 2006), are also controlled by synoptic-scale meteorology (Sun et al., 2014; Zheng et al., 2015). Therefore, the influence of synoptic condition on the multi-day PM pollution need to be considered together with the local and regional sources.

In the late February 2014, South Korea experienced severe multi-day haze (Shin et al., 2014; Park et al., 2015; Kim et al., 2016). We collected daily $PM_{2.5}$ samples at both an urban site in Seoul and an upwind background site in Deokjeok Island over the Yellow Sea, during the haze and following clean periods. Because PM in Seoul is significantly affected by the local sources while PM in Deokjeok Island is mostly attributed to the regional sources, the comparison between two sites is our approach for the source appointment. In this work, we explored a combined effect of local emissions, regional transport, and synoptic
conditions on the prolonged fine particulate pollution in the Asian continental outflow region. Chemical compositions including inorganic and carbonaceous species of $PM_{2.5}$ at both sites during the haze and clean periods were characterized. In addition, the temporal evolution of the haze episode was examined with changes in chemical components and meteorological factors. Finally, relevant synoptic conditions that induced the multi-day haze event are further discussed.

## 2 Measurement and data

### 2.1 Sampling overview

Daily $PM_{2.5}$ and $PM_{10}$ samplings were conducted at the urban site located inside the Korea Institute of Science and Technology (KIST) in the northeastern Seoul (37.603°N, 127.047°E, 58 m above sea level) and the background site located on Deokjeok Island over the Yellow Sea (37.233°N, 126.149°E, 185 m above sea level) between February 23 and March 9 in 2014 (Fig. 1a). The sampling period includes the multi-day haze pollution in late February and the following clean-air period in early
March. The KIST site is located downwind of the downtown core and surrounded by a small urban forest (about 60 ha) and residential area. Thus the measurement at this site represents the particulate air quality of Seoul influenced by both local and regional sources. The Deokjeok site located on the offshore island is about 90 km to the west-southwest from the KIST site in





Seoul. Since significant emission sources do not exist in the island, the measurement at Deokjeok site mainly represents the regional sources.

The daily sampling started from 0900 local time (LT) and was carried out for 24 h on each day. The $PM_{2.5}$ samples for the mass measurement and the ionic analysis were collected on pre-weighted 47 mm Teflon filters (Zefluor™, Pall Corp., Port

Washington, NY) by using a Teflon-coated aluminum cyclone with a cut size of 2.5 μm and flow rate of 16.7 L min$^{-1}$ (URG Corp., Chapel Hill, NC). PM with diameter of 10 μm or less ($PM_{10}$) was also collected in the same way to compare the mass concentration with $PM_{2.5}$. The $PM_{2.5}$ sample for carbonaceous and organic compounds analyses is collected on 203 mm × 254 mm quartz fiber filters (Whatman Inc., Maidstone, UK) by using a high-volume air sampler with flow rate of 1000 L min$^{-1}$ (Andersen Instruments Inc., Atlanta, GA). All samples were sealed immediately after the collection and stored in freezers to

prevent possible contamination.

**2.2 Analytical procedure**

The mass concentrations of $PM_{2.5}$ and $PM_{10}$ were measured by the Mettler MT5 microbalance (Mettler-Toledo, Greifensee, Switzerland) after 24 h standing of the Teflon filter sample in a desiccator. Then, the sample was sonicated in a mixture of 0.5 mL of ethanol and 14.5 mL of distilled/deionized water for 30 min to measure water-soluble ion concentrations in $PM_{2.5}$.

Concentrations of sulfate ($SO_4^{2-}$), nitrate ($NO_3^-$), chloride ($Cl^-$), ammonium ($NH_4^+$), potassium ($K^+$), calcium ($Ca^{2+}$), sodium ($Na^+$), and magnesium ($Mg^{2+}$) were measured by the Dionex 2000i/SP ion chromatograph (Dionex, Sunnyvale, CA). To analyze organic carbon (OC) and element carbon (EC) in $PM_{2.5}$, a piece of the quartz fiber filter (10 mm × 15 mm) was used with a thermal/optical carbon aerosol analyzer (Sunset Laboratory, Tigard, OR) based on the National Institute for Occupational Safety and Health (NIOSH) method 5040 (Birch and Cary, 1996).

To identify and measure concentrations of individual organic compounds, one-half of the quartz fiber filter was used and ultrasonicated twice in a mixture of dichloromethane and methanol (3:1; v/v) for twice 30 min. The filter composite was spiked with 11 isotopically-labeled surrogated standards and blown down to 100 mL with a Zymark TurboVap 500 concentrator (Zymark Corporation, Hopkinton, MA) under pure nitrogen stream at 40°C. The extract was filtered by syringe with PTFE membrane filter (ID 25 mm, pore size 0.45 μm, Pall Corporation, New York) and further reduced by gentle solvent evaporation

with a stream of high purity nitrogen to a final volume of 0.5 ± 0.1 mL. The GC-MS analysis was carried out on a Hewlett Packard 7890A gas chromatograph coupled to a 5975C mass selective detector (Agilent, Palo Alto, CA) in the synchronous selected ion monitoring (SIM)/Scan mode. A 1 μL sample was injected on splitless mode at 240°C. The mass spectrometer was operated on electron impact (EI) mode at 70 eV and scanned from 40 Da to 550 Da at the source temperature of 230°C.

The organic compounds were classified into five groups such as *n*-alkanes, polycyclic aromatic hydrocarbons (PAHs),

monocarboxylic acids, sugars, and dicarboxylic acids. These are seventeen *n*-alkanes: $C_{20}$–$C_{36}$; fifteen PAHs: phenanthrene (PHE), anthracene (ANT), fluoranthene (FLA), pyrene (PYR), benz[a]anthracene (BaA), chrysene (CHR), benzo[b]fluoranthene (BbF), benzo[e]pyrene (BeP), benzo[a]pyrene (BaP), perylene (PER), 1,3,5-triphenylbenzene (TPB), indeno[1,2,3-cd]pyrene (IcdP), dibenz[a,h]anthracene (DahA), benzo[ghi]perylene (BghiP), and coronene (COR); nineteen



monocarboxylic acids: $C_6$–$C_{20}$ including oleic or elaidic acids ($C_{18:1}$) and *cis*-pinonic acid; nine sugars: arabinose, ribose, levoglucosan, xylose, fructose, mannose, galactose, glucose, and sucrose; nineteen dicarboxylic acids: malonic acid, methylmalonic acid, maleic acid, methylmaleic acid, succinic acid, methylsuccinic acid, fumaric acid, glutaric acid, 2-methylglutaric acid, D-malic acid, adipic acid, pimelic acid, phthalic acid, suberic acid, iso-phthalic acid, tere-phthalic acid, azelaic acid, sebacic acid, and undecanedioic acid. Detailed description of aforementioned chemical analyses can be found in Choi et al. (2016).

## 2.3 Data

The Korean Ministry of Environment (KMOE) provides 1 h average concentrations of $PM_{10}$, $SO_2$, $NO_2$, CO, and ozone ($O_3$) at 257 urban sites, 19 suburban sites, and 3 background sites over South Korea (NIER, 2016). The measurements of each species are conducted by β-ray absorption method for $PM_{10}$, pulse ultraviolet fluorescence method for $SO_2$, chemiluminescent method for $NO_2$, non-dispersive infrared method for CO, and ultraviolet photometric method for $O_3$. In the present study, we employ hourly data of $PM_{10}$ and other gaseous species measured at the Seongbuk site (1.7 km northwest from the PM-sampling site inside the KIST) in Seoul and the Deokjeok site (right next to the PM-sampling site) to analyze and compare with the sampling data (Fig. 1a). As shown in Fig. 1b, the mass concentration of $PM_{10}$ from daily filter samples in Seoul (KIST) and Deokjeok are consistent with the daily $PM_{10}$ concentrations measured by β-ray absorption in Seoul (Seongbuk) and Deokjeok ($r^2 = 0.99$ in Seoul and $r^2 = 0.97$ in Deokjeok). Also we select 247 sites based on data availability for the analysis period to explore the changes in spatial distribution of $PM_{10}$ over South Korea during the prolonged haze episode. Daily $PM_{2.5}$ concentrations in five Chinese cities of Beijing, Tianjin, Dalian, Jinan, and Yantai were obtained from a public weather website (http://www.tianqihoubao.com/aqi/).

Hourly meteorological variables such as temperature, RH, wind speed observed at the Seoul weather station (8 km southwest from the KIST site) provided by the Korea Meteorological Administration (KMA) are additionally used in this study (Fig. 1a). To investigate boundary layer height in Seoul, the European Centre for Medium-Range Weather Forecasts Reanalysis Interim (ERA-Interim) data at a grid point within Seoul (37.5°N, 127.0°E, 12.3 km south-southwest from the KIST site) are employed. To investigate synoptic condition during the haze period, geopotential height and wind fields at 850 hPa and 500 hPa derived from the ERA-Interim data together with aerosol optical depth (AOD) at 550 nm from the Moderate Resolution Imaging Spectroradiometer (MODIS) onboard the Terra and Aqua satellites are used.

## 3 Results and Discussions

### 3.1 Multi-day haze episode in Seoul

In the late February 2014, daily $PM_{10}$ concentration in Seoul (by β-ray absorption) first exceeded the World Health Organization (WHO) 24 h mean guideline of 50 µg m$^{-3}$ (WHO, 2006) on February 20, increased over the South Korean 24 h mean standard of 100 µg m$^{-3}$ (NIER, 2016) from February 23 to 28, and fell back below 50 µg m$^{-3}$ on March 2. Then $PM_{10}$



concentration higher than 50 μg m$^{-3}$ was again recorded from March 3 to 4 (Fig. 1b). Daily PM$_{2.5}$ concentration in Seoul (by filter sampling) shows a similar temporal pattern. The PM$_{2.5}$ mass concentration in Seoul recorded its highest value of 148 μg m$^{-3}$ on February 24 and stayed near 100 μg m$^{-3}$ until February 28 (Fig. 1b). The measured PM$_{2.5}$ in Seoul exceeds not only the WHO 24 h mean guideline of 25 μg m$^{-3}$ (WHO, 2006) but also the South Korean 24 h mean standard of 50 μg m$^{-3}$ (NIER, 2016) for 9 days between February 23 and March 4, except March 2. In Deokjeok, the high concentration of PM$_{2.5}$ near or higher than 100 μg m$^{-3}$ continued until February 26, but its PM$_{2.5}$ level started to fall down on February 27, when the PM$_{2.5}$ level was still high in Seoul (Fig. 1b).

The PM$_{2.5}$ mass concentration in Deokjeok was generally 70–90% of that in Seoul throughout the measurement period. It was, however, slightly higher than that in Seoul on a day in the mid-stage of the haze (February 26) and then showed only 40–60% of that in Seoul during the following late stage of the episode (February 27–March 1). A similar trend was reported in five upwind Chinese cities across the Yellow Sea, which also experienced the high PM$_{2.5}$ concentrations from February 20 to 26 and rapid drop of the PM$_{2.5}$ levels on February 27 (Fig. 1c).

The wind direction during the overall analysis period was mostly westerly or west-northwesterly, except easterly winds on February 26. Therefore, the high PM$_{2.5}$ level in Deokjeok is most likely due to the regional transport from China, evidenced by the high PM$_{2.5}$ levels in the upwind Chinese cities and negligible local emissions in and near the Deokjeok Island. On the other hand, the high PM$_{2.5}$ level in Deokjeok on February 26 seems to result from the easterly transport of pollutants from the SMA. The high PM$_{2.5}$ levels in Seoul but the low PM$_{2.5}$ concentrations in Deokjeok on the following 3 days show that the prolonged haze period in Seoul is not just a result of transboundary transport of pollutant.

The durable haze widely affected PM levels over South Korea as represented in the spatial distribution of PM$_{10}$ concentrations (Fig. 2). The obvious western-high/eastern-low concentration pattern appears during the haze enhancement (February 22–24), but such westward gradient of PM$_{10}$ levels becomes weaker after the appearance of nationwide high concentration (February 25–27). PM$_{10}$ concentrations finally decrease but still remain relatively high in the SMA and inland area in the last stage of the prolonged haze (February 28–March 1). The westward PM$_{10}$ gradient with the high PM concentrations at Deokjeok (Fig. 1b) before February 26 supports the regional influence on the early stage of the haze period. On the other hand, the weakened westward PM$_{10}$ gradient, the high PM$_{10}$ levels in the SMA and inland area, and the low PM concentrations at Deokjeok (Fig. 1b) after February 26 imply a local influence on the final stage of the haze period in Seoul.

### 3.2 Formation of PM$_{2.5}$ in Seoul and background

Chemical speciation of PM$_{2.5}$ and gaseous pollutants in Seoul and Deokjeok during the haze and clean period reveal spatial dynamics of PM$_{2.5}$ influenced by urban and its upwind background in the downwind of East Asia. The average chemical compositions of PM$_{2.5}$ in both Seoul and Deokjeok for the haze period (February 23–28) and following clean period (March 5–9) are presented in Fig. 3 and Tables 1–2. Different characteristics of PM$_{2.5}$ in between downwind urban and upwind background are revealed from the comparisons of PM$_{2.5}$ compositions between Seoul and Deokjeok and summarized in Table 3. PM$_{2.5}$ properties during the haze and clean periods are compared and characterized in Table 4.



### 3.2.1 Local meteorological conditions in Seoul

Seoul experienced warm, humid, and stagnant conditions within the shallow boundary layer during the haze period (Table 1 and Figs. 4e–f). The decrease of boundary layer height and low wind speed effectively interrupt vertical mixing and increases PM concentrations near the ground by the accumulation of primary aerosols and secondary aerosol precursors (Zheng et al., 2015). As a result of the high PM concentrations, daytime visibility in Seoul was significantly reduced to its minimum value of 1.3 km in the morning on February 25. As mentioned in the previous section, wind directions were mostly westerly or west-northwesterly over the measurement period, except easterly only on February 26. Therefore, the regional influences on the severe haze can be inferred from the high concentration of each chemical component during the haze period in both Seoul and Deokjeok. On the other hand, the local contribution to the long-lasting haze in Seoul is clearly shown by the tendency of the lower mass concentration of each component in Deokjeok than that in Seoul (Table 1).

### 3.2.2 Gaseous species in Seoul and background

Among the four gas species investigated in this study, $SO_2$ and $NO_2$ are important species closely related to sulfate and nitrate aerosols. The higher $SO_2$ concentration in the haze period in Deokjeok than that in Seoul (Table 1), and much lower emission intensity of $SO_2$ in the SMA (3.1 t km$^{-2}$) than that in the Chinese Yellow Sea coastal provinces (Li et al., 2017; NIER, 2015) provide evidence that $SO_2$ in Seoul was greatly influenced by regional transport from China. On the other hand, significantly higher concentration of $NO_2$ in Seoul than in Deokjeok not only during the haze period but also during the clean period (Table 1) is attributed to the high $NO_x$ emission in the SMA (23.3 t km$^{-2}$) from vehicles and industrial combustion sources (NIER, 2015).

The high $NO_x$ emission in the SMA is presumably a major factor for different $O_3$ levels in between Seoul and Deokjeok. In general, $O_3$ level in Seoul is lower than that in Deokjeok due to titration effects by substantial emissions of $NO_x$ in the SMA (Seo et al., 2014). In Seoul, the haze period $O_3$ level is lower than the clean period one (Table 1) due to the enhanced titration in high $NO_x$ conditions. However, Deokjeok shows higher $O_3$ concentration in the haze period than in the clean period (Table 1). This higher $O_3$ level during the haze may result from the enhanced $O_3$ production due to the slightly increased $NO_2$ concentration by the regional transport (Lee et al., 2014) and/or the transport of $O_3$ itself (Oh et al., 2010; Seo et al., 2014). CO, which is an incomplete combustion product of fossil fuels or biomass/biofuels, is affected by both local emissions in the SMA and the regional transport from China and thus shows similar characteristics to total $PM_{2.5}$; higher concentration in Seoul than in Deokjeok and also in the haze period than in the clean period (Table 1).

### 3.2.3 Local and regional formation of $PM_{2.5}$ during the haze period

In the haze period, $PM_{2.5}/PM_{10}$ ratios in Seoul and Deokjeok are high (~ 0.8) (Table 1), which shows dominance of the anthropogenic fine-mode aerosols (Kim et al., 2007). Slightly larger $PM_{2.5}/PM_{10}$ ratios in Deokjeok compared to those in Seoul (Table 1) suggest more influence of the secondary aerosol formation in the long-range transport (Irei et al., 2015) as well as



less coarse particle sources such as non-exhaust vehicle emissions, construction, fugitive soil and dust resuspended by road traffic, and industry (Kassomenos et al., 2012).

The prolonged haze episode is characterized by large proportion of secondary aerosols in $PM_{2.5}$ (Fig. 3). The average fractions of secondary inorganic aerosol (SIA) species (sulfate, nitrate, and ammonium) in total $PM_{2.5}$ mass during the haze period are

76% in Seoul and 65% in Deokjeok, while those during the clean period are reduced to 47% in Seoul and 51% in Deokjeok (Fig. 3 and Table 1). The sulfur oxidation ratio (SOR = $n$ $SO_4^{2-}$ / [$n$ $SO_4^{2-}$ + $n$ $SO_2$]) ($n$ refers to the molar concentration) and the nitrogen oxidation ratio (NOR = $n$ $NO_3^-$ / [$n$ $NO_3^-$ + $n$ $NO_2$]) represent the atmospheric conversion of precursor gases to SIA through the oxidation and partitioning (Squizzato et al., 2013). These two ratios are also higher during the haze period than those during the clean period in Seoul (Table 1).

During the clean period, the SIA fraction in Seoul (47%) is slightly smaller than that in Deokjeok (57%) owing to the smaller sulfate fraction in Seoul (18%) than that in Deokjeok (26%). During the haze period, on the other hand, the SIA fraction in Seoul (76%) is much larger than that in Deokjeok (65%) mainly due to the larger nitrate fraction in Seoul (28%) than that in Deokjeok (13%) (Table 1). The large fraction of nitrate and local emission on $NO_2$ in Seoul indicate that a majority of nitrate aerosols is locally produced in the SMA. And the large fraction of sulfate and dominant influences of regional transport on

$SO_2$ in Deokjeok indicate that sulfate aerosols are formed secondarily during the long-range transport.

The high OC/EC ratios (> 7) during the haze period in both Seoul and Deokjeok (Table 1) indicate the large proportion of secondary organic aerosols (SOA) in $PM_{2.5}$ since EC is a representative primary constituent while OC has both primary and secondary sources (Turpin and Huntzicker, 1995). As shown in Table 1, EC concentrations in Seoul during both haze and clean periods are higher than those in Deokjeok. This shows that EC in Seoul is largely contributed by local emissions while that in Deokjeok is mostly influenced by regional transport. The OC/EC ratio is higher in Deokjeok than in Seoul during both

haze and clean periods. This implies larger SOA fraction in carbonaceous $PM_{2.5}$ in the background compared to Seoul probably due to the secondary production during the long-range transport, as discussed in Sect. 3.3.

### 3.3 Organic components in $PM_{2.5}$

In the present study, five organic compound groups ($n$-alkanes, PAHs, monocarboxylic acids, sugars, and dicarboxylic acids)

were found and analyzed. Various diagnostic ratios from these individual organic compounds are useful markers for identification of primary anthropogenic and biogenic sources as well as secondary formation and aging of organic aerosols (OA). The average concentrations and diagnostic ratios of the organic compounds are presented in Table 2. Different source characteristics between Seoul and upwind background and between the haze and clean periods derived from Table 2 are summarized and added in Tables 3 and 4.

### 3.3.1 Emission sources



PAHs are combustion byproducts of all types of organic matters, especially related to the incomplete combustion of fossil fuels and biomass burning (Nisbet and LaGoy, 1992; Bi et al., 2003). Various diagnostic ratios with individual PAH species are helpful to search their emission sources (Tobiszewski and Namieśnik, 2012).

Prior studies reported that FLA/(FLA + PYR) is smaller than 0.4 for petrogenic source, within the range of 0.4–0.5 for the fossil fuel combustion source, and larger than 0.5 for grass, wood, coal combustion source (De La Torre-Roche et al., 2009). The high values of FLA/(FLA + PYR) in both Seoul (~ 0.55) and Deokjeok (~ 0.60) (Table 2) indicate biomass and coal combustion sources but more influence of fossil fuel combustion in Seoul. BaA/(BaA + CHR) is smaller than 0.2 for petrogenic sources, within the range of 0.2–0.35 for the coal combustion sources, and larger than 0.35 for the vehicular emissions (Yunker et al., 2002; Akyüz and Çabuk, 2010). The values of BaA/(BaA + CHR) in both Seoul (0.26–0.28) and Deokjeok (~ 0.22) (Table 2) imply coal combustion sources, but the higher ratios in Seoul than Deokjeok also indicate more influence of vehicles. ANT/(ANT + PHE) is smaller than 0.1 for petrogenic source but larger than 0.1 for pyrogenic source (Pies et al., 2008). ANT/(ANT + PHE) ratios in Table 2 show more pyrogenic-like source in both Seoul and Deokjeok (~ 0.1) but petrogenic source in Deokjeok especially during the haze (~ 0.06). IcdP/(IcdP + BghiP) is lower than 0.2 for petrogenic source, within the range of 0.2–0.5 for petroleum combustion source, and higher than 0.5 for grass, wood, and coal combustion source (Yunker et al., 2002). In Table 2, the high IcdP/(IcdP + BghiP) ratios during the haze (~ 0.6) indicate biomass and coal combustion sources while the low ratios during the clean period (< 0.5) implies petroleum combustion source.

Sugars mainly originate from biomass burning and numerous primary biological aerosols (Bi et al., 2008; Fu et al., 2012). Especially, levoglucosan, which comprises more than 90% of the total sugars, is produced by pyrolysis of cellulose and regarded as a representative marker for biomass burning (Simoneit et al., 1999). Similar to EC, the high levoglucosan concentration in Seoul, even higher than the haze-period levoglucosan levels in Deokjeok, implies that Seoul is largely affected by local biomass burning together with regional transport.

$n$-Alkanes have biogenic sources including particles shed from the epicuticular waxes of vascular plants as well as anthropogenic sources including fossil fuel and biomass combustion (Simoneit, 1991). Since $n$-alkanes of recent biogenic origin show a distinct odd carbon number preference (Simoneit, 1991; Rogge et al., 1993), the carbon preference index (CPI$_{odd}$) defined by a concentration ratio of odd-to-even numbered homologues is higher than 3 for more biogenic sources while that is close to 1 for more anthropogenic combustion sources (Simoneit, 1989). Together with the high percentage of plant wax $n$-alkanes (Wax $C_n = [C_n] - ([C_{n+1}] + [C_{n-1}]) / 2$), the high CPI$_{odd}$ in Seoul during the haze reflects more biogenic VOCs in Seoul from the urban forest. More biogenic emissions in Seoul than in Deokjeok are also supported by higher concentrations of $cis$-pinonic acid, which is a representative biogenic SOA species that originate from the O$_3$ reactions of monoterpene (Zhang et al., 2010). It should be noted that only high molecular weight $n$-alkanes (C$_{20}$–C$_{36}$) were analyzed in this work to focus on biogenic contributions.

To sum up with the above diagnostics, both Seoul and Deokjeok are commonly influenced by biomass burning and coal combustion sources. However, Seoul is more affected by petroleum combustion and vehicular emissions especially during the





haze, while Deokjeok is more affected by petrogenic-like sources. Seoul seems to also have local biomass burning and biogenic emission sources and can be affected by these sources during the haze.

### 3.3.2 Secondary formation and aging process

Dicarboxylic acids originate not only from the primary sources like fossil fuel combustion and biomass burning but also the
secondary sources like gas-particle partitioning of semivolatile products from the photooxidation of anthropogenic or biogenic precursors (Rogge et al., 1993; Zhang et al., 2010) and aqueous chemistry in aerosol waters (Zhang et al., 2016). In Table 2, the large proportions of dicarboxylic acid carbon in OC and their high oxygen-to-carbon ratios (O:C) during the haze period support that the SOA has dominantly formed through oxidation in the stagnant condition. More dicarboxylic acids (carbon basis) in OC and their higher O:C ratio in Deokjeok than those in Seoul are consistent with conclusions from the
aforementioned high OC/EC ratio in Deokjeok (Table 1) in Sect. 3.2.3: OA in the upwind background shows more influence of chemical aging and secondary production during the long-range transport, but those in Seoul originate from a mixture of transported secondary sources and local primary sources.

The oxidation and aging of the aerosols could be further supported by diagnostic ratios of PAH isomers and $C_{18}$ monocarboxylic acids. Benzo[a]pyrene (BaP) is photodegraded more rapidly than its PAH isomer, benzo[e]pyrene (BeP), and
thus BaP/(BaP + BeP) is close to 0.5 for the fresh particles while it is smaller than 0.5 for the aged particles (Oliveira et al., 2011; Tobiszewski and Namieśnik, 2012). In Table 2, smaller BaP/(BaP + BeP) ratios during the haze period than during the clean period show more aging processes during the haze period. Monocarboxylic acids (or fatty acids) originate from the fossil fuel combustions, biomass burning, meat cooking, and biogenic sources such as epicuticular plant waxes (Cheng et al., 2004; Choi et al., 2016). Among the monocarboxylic acids, the ratio of saturated stearic acid ($C_{18:0}$) to unsaturated oleic and elaidic
acids ($C_{18:1}$) is higher for more aged aerosols because the unsaturated monocarboxylic acids are rapidly degraded to fragments via heterogeneous oxidation (Simoneit et al., 1991; Cheng et al., 2004). Higher $C_{18:0}/C_{18:1}$ with smaller BaP/(BaP + BeP) in Deokjeok than in Seoul (Table 2) indicate that OA in Seoul are fresher and those in Deokjeok are more aged.

### 3.4 Temporal evolution of the multi-day haze

Figures 4 and 5 show the temporal evolution of the concentrations of SIA species and their precursor gases, carbonaceous
aerosol components and CO, and the five individual organic compound groups measured at both Seoul and Deokjeok sites, together with the meteorological factors in Seoul. Correlations between meteorological parameters and chemical components in Seoul are listed in Table 5 to help understanding the role of meteorological factors in the temporal variation of each component.

Sulfate profiles (Fig. 4a) roughly follow the trend of $PM_{2.5}$ concentrations that is higher in Seoul than in Deokjeok (Fig. 1b).
A rapid increase of sulfate concentrations between February 23 and 24 in Seoul indicates the introduction of sulfate by regional transport. Similar concentrations of sulfate in Seoul and Deokjeok peaking on February 24 and decaying afterwards confirm a strong impact of regional transport on sulfate in the early stage of the haze. A sharp increase of specific humidity on February



24 shows abrupt changes in the air mass property and thus source origin (Fig. 4e). The sulfate concentrations in Deokjeok gradually decreased between February 25 and 28, while those in Seoul stayed 30 µg m$^{-3}$. Thus the prolonged high sulfate level in Seoul must be affected by local production. Increasing SOR in Seoul despite the decreasing SOR in Deokjeok between February 26 and 28 supports growing local sulfate formation in Seoul during the late stage of the multi-day haze (Fig 4b). The

warm and humid air conditions during the haze period (Fig. 4e) could be also conducive to both gas-phase and aqueous-phase oxidation of $SO_2$ (Liang and Jacobson, 1999; Seinfeld and Pandis, 2006), as shown in Table 5 by the high correlation coefficient between SOR and temperature ($r = 0.72$) or RH ($r = 0.59$). Interestingly, the correlation of sulfate with temperature ($r = 0.64$) or RH ($r = 0.47$) are weaker than those of SOR, due to transport of external sulfate into Seoul during the early stage of the haze.

As illustrated in the previous section, both $NO_2$ and nitrate profiles show much higher concentrations in Seoul than those in Deokjeok owing to large $NO_x$ emission in Seoul (Fig. 4c). The high dependence of $NO_2$ and nitrate on the boundary layer and winds (Table 5) in Seoul reflects significant influences of such meteorological factors on the species originated from local primary emissions and secondary production. For example, $NO_2$ concentration in Seoul gradually increased with lowering of boundary layer height until February 25 and then started to decrease on February 26 with rising of boundary layer height (Fig.

4f). The high wind speeds on February 27 and March 2 are corresponding to the decreasing peaks of $NO_2$ and nitrate concentrations on the same days. In addition, the high $NO_2$ and nitrate concentrations in Deokjeok on February 26 must be affected by transport from the SMA since the easterly winds were observed only on that day as referred in Sect. 3.1. In terms of nitrate aerosol formation in Seoul, an important role of aqueous chemistry could be inferred from the high correlations of RH with nitrate ($r = 0.53$) and NOR ($r = 0.67$) (Table 5). A recent study using smog chamber experiments and measurement

data (Lim et al., in preparation, 2017) reveals that $NO_x$ photochemistry under the high RH condition facilitates hygroscopic growth of aerosols through water and $HNO_3$ uptake cycles and thus contributes to the high nitrate proportion to $PM_{2.5}$ in Seoul (Fig. 3a).

Temporal evolution of CO shows mixed characteristics of $SO_2$ and $NO_2$ profiles (Fig. 4d). Like $SO_2$ concentrations, CO levels gradually increased in both Seoul and Deokjeok in the early stage of the haze, and this shows continuous regional influences

during the haze period. On the other hand, the high concentration difference in CO during the haze period between Seoul and Deokjeok as well as the small concentration difference in CO on the high wind speed days are the $NO_2$ characteristics that indicate local influence on the high CO levels in Seoul. EC and OC generally show similar temporal variation to CO (Fig. 4d) that are high-negatively correlated to wind speed and boundary layer height (Table 5).

*n*-Alkanes, PAHs, monocarboxylic acids, and sugars commonly show the high concentrations in Seoul during the haze period

and the low concentration peaks on high wind days (February 27 and March 2), and the high concentration peaks in Deokjeok on the easterly wind day (February 26) (Figs. 5a–d). These are the locally originated characteristics as shown in $NO_2$. Higher concentrations of *n*-alkanes, monocarboxylic acids, and sugars during the haze period than those during the clean period in Deokjeok show regionally transported characteristics of $SO_2$. PAHs in Deokjeok show low concentrations with small temporal variability, except high concentration peaks on February 26 associated with transport from the SMA by easterly winds (Fig.





5b). And this clearly indicate local influence of primary organic compounds on the high OC levels in Seoul during the prolonged haze.

As briefly mentioned in Sect. 3.3, dicarboxylic acids have both primary and secondary sources. Concentration differences for dicarboxylic acids in Seoul and Deokjeok are small during the clean period, but large during the haze period (except on February 26) probably due to influences of local primary emissions and secondary formation in Seoul (Fig. 5e). On the other hand, high concentrations of dicarboxylic acids in Deokjeok during the early stage of the prolonged haze must be due to the regional transport of pre-formed products (Fig. 5e). The O:C ratio of dicarboxylic acids that stayed high during both early and late stages of the haze period in both two places indicates secondary production as a source of the organic compounds (Fig. 5f). To sum these up, the high concentration of dicarboxylic acids in Seoul during the multi-day haze is a combined result from local emissions, secondary formation, and early stage regional transport.

It should be noted that significant correlations with RH were also found for dicarboxylic O:C ratio ($r = 0.54$) and OC/EC ratio ($r = 0.51$) (Table 5). The stronger and more significant correlation of RH with OC/EC ratio than that with the OC concentration ($r = 0.33$) suggests probable aqueous-phase processes for the SOA formation during the prolonged haze. This is also supported by the much stronger correlations of RH with dicarboxylic acids ($r = 0.53$) and their O:C ratio ($r = 0.54$) than those with other primary organic compounds ($0.25 < r < 0.38$) (Table 5).

### 3.5 Role of synoptic condition on the multi-day haze

Local meteorological factors, which play an important role in accumulation of both primary pollutants and secondary aerosol precursors as well as the secondary formation processes, are largely controlled by synoptic-scale conditions (Zheng et al., 2015). In addition, some specific distributions of high- and low-pressure systems could provide a favorable pathway to transport external pollutants (Lee et al., 2011; Oh et al., 2015). We examined the influence of synoptic conditions on the day-to-day variation of meteorological factors and temporal evolution of the multi-day haze using geopotential height and wind fields at 850 hPa (about 1.5 km altitude) together with AOD.

From February 19 to 21, a migratory anticyclone developed over southern China was moving eastward from the Sichuan Basin to the Yangtze River Delta, and the high AOD was observed in the North China Plain (Fig. 6). As the high-pressure system moved eastward, a clockwise circulation transported aerosols, which had been accumulated in the stable and stagnant air mass, into the North China and therefore resulted in the rapid rising $PM_{2.5}$ concentrations in this region (Fig. 1c). This high-pressure system slowed down and stayed over the East China Sea on between February 22 and 24. During the period, the Korean Peninsula was under the direct influence of the slow anticyclonic circulation along the northwestern flank of the stagnant high-pressure system. The anticyclonic flow induced warm and humid advection along the eastern coast of China and accumulated and produced primary and secondary aerosols over the North China and the Yellow Sea, as shown by the high AOD values over the regions. Backward trajectories passing through these regions indicate dominant influence of regional transport on the nationwide increase of PM concentrations in South Korea in the early stage of the prolonged haze (Figs. 1b and 2b–d).





As a trough develops over southern China on February 25, the northwestern flank of the anticyclone was elongated and stretched toward the Korean Peninsula. This pattern provided a stable condition over South Korea, but also caused weak transport from the west. The anticyclone started to move eastward again on February 26, and following low-pressure system resulted in easterly winds in the SMA. The easterly winds induced a warm peak of temperature and a slightly decreasing peak of specific humidity in Seoul (Fig. 4e), related to foehn-related phenomena caused by a north-south mountain range in the east of the Korean Peninsula. Staggered backward trajectories over/near the Korean Peninsula until March 1 imply a dominant role of local emissions and production in the late stage of the haze episode. The synoptic conditions between March 2 and 3 resemble those between February 20 and 25, however, high-pressure system was located a little bit northward and quickly moved eastward. Aerosols were therefore less accumulated over the North China (Fig. 1c) and quickly passed through the Korean Peninsula between March 3 and 4 (Fig. 1b). The high concentrations of $SO_2$, sulfate, and nitrate in both Seoul and Deokjeok but very low concentration of $NO_2$ in Deokjeok (Figs. 4a and 4c) support that the high $PM_{2.5}$ levels on these two days were highly affected by quick transport from China.

The haze-related high-pressure system, which was slowly developed over China and moved into the Korean Peninsula, is analogous to composite anomaly pattern of 850 hPa geopotential height for the multi-day high $PM_{10}$ episodes in Seoul reported by Oh et al. (2015). However, as shown by the difference between two subcases (February 20–March 1 and March 3–4), duration of the haze episode and their sources (regional or local sources) also depends on zonal speed of the synoptic-scale systems. One key weather pattern to hinder the eastward advance of weather systems in the midlatitudes is atmospheric blocking, which usually has a dipole structure of equatorward cyclone and poleward anticyclone or an omega ($\Omega$)-shaped ridge (Pelly and Hoskins, 2003; Tyrlis and Hoskins, 2008). In the geopotential height fields at 500 hPa (about 5.5 km altitude), a planetary wave ridge appeared over the Gulf of Alaska on February 21 (Fig. 7). The ridge developed into the omega-shaped blocking over Alaska (February 24–March 1) and was responsible for the stagnant synoptic-scale weather systems during the period by interrupting zonal circulation. In contrast, the zonal propagation of synoptic disturbances became fast after dissipation of the blocking anticyclone (March 2–5), and as a result, the fast-moving high-pressure system and following low-pressure system induced the quick transport of external pollutants but less accumulation of domestic pollutants in South Korea on March 3–4.

## 4 Conclusions

In the present study, the evolution of the multi-day haze in late February 2014 in the highly industrialized region located downwind of the Asian continental outflow has been investigated by $PM_{2.5}$ chemical speciation in Seoul and its upwind background, Deokjeok Island, over the Yellow Sea. $PM_{2.5}$ in Seoul is different from those in Deokjeok, showing higher concentrations of primary aerosols, larger nitrate fraction, more biogenic VOCs and precursors, and more pyrogenic sources including fossil fuel combustion and biomass burning, but smaller sulfate and less-oxidized SOA fractions. Such differences




reflect that $PM_{2.5}$ in Seoul is affected by both local and regional sources while that in Deokjeok is affected by the regional transport.

During the haze period, $PM_{2.5}$ level in Seoul is largely increased by primary and secondary aerosols. In particular, SIA species occupy 76% of total $PM_{2.5}$ mass, and OC/EC ratio reaches 7.29. The increase in primary and secondary aerosols are closely related to the warm, humid, and stagnant meteorological condition during the haze period, which is conducive to the accumulation of pollutants and the oxidation of precursors. The high correlations of RH with nitrate, NOR, SOR, OC/EC ratio, and dicarboxylic O:C ratio show an important role of aqueous processes on the oxidation and formation of the secondary aerosol species.

Temporal evolution of the $PM_{2.5}$ chemical components in both places shows a sequential influence of regional and local sources on the prolonged haze period. High concentrations of sulfate in both Seoul and Deokjeok in the early stage of the haze episode indicate a significant regional influence on the high sulfate fraction in Seoul (30%) especially at the onset of the multi-day haze. On the other hand, much higher concentration of nitrate in Seoul (28% of $PM_{2.5}$) than that in Deokjeok for the overall episode shows local formation and accumulation under the influence of a low boundary layer height. Concentrations of organic compounds show similar behaviors to the nitrate concentration. Stagnant atmospheric conditions extend the haze period by local contributions.

Transboundary transport of external pollutants and day-to-day variation of the local meteorological factors are controlled by synoptic-scale condition. Migratory anticyclone in southern China accumulates aerosols over the North China Plain and the Yellow Sea. As the high-pressure system slowly moves eastward, weak anticyclonic flow transports the accumulated aerosols into the Korean Peninsula in the early stage of the multi-day haze. In the mid- and late stages, the eastward-moving high-pressure system governs the Korean Peninsula and provides favorable meteorological conditions to accumulation and local formation of aerosols. Duration of the haze depends on zonal propagation speed of the synoptic-scale systems. During the long-lasting haze period, a blocking ridge developed over Alaska hinders zonal flows and the eastward migration of the anticyclone slows down over the East China Sea.

Our findings show an important role of synoptic condition on the onset and evolution of the multi-day haze in the downwind region of East Asia with explicit chemical details of haze development sequentially by regional transport and local sources. Since most of the megacities in the populated and industrialized region act as both importer and exporter of air pollutants, its air quality must be affected by both regional and local emission sources. As a synoptic system conducive to accumulation of air pollutants and production of secondary aerosols moves from the upwind region, air quality in the downwind region is affected not only by the transport of air pollutants from the upwind area but also by the propagation of such conducive weather system itself. Although the multi-day high PM episode started with the transport of external pollutants, local emission in the downwind region under certain meteorological conditions exacerbates air quality and prolongs the haze period. For instance, the high PM concentration in Deokjeok, where the local emissions are negligible, decreased 2–3 days earlier than that in Seoul in the multi-day haze episode in this study. Duration of the pollution episode also depends on the zonal propagation speed of such haze-favorable weather system, and in particular, the atmospheric blocking pattern prolongs the haze period.





**Acknowledgements**

This work was supported by Korea Institute of Science and Technology (KIST) and by the National Research Foundation of Korea grant NRF–2011–0028597. Yong Bin Lim acknowledges the support from Brainpool Fellowship by Ministry of Science, ICT and Future Planning, South Korea (152S-5-2-1416).

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



**Table 1: The average and standard deviation of PM$_{2.5}$ chemical compositions, related gas concentrations, and meteorological factors in Seoul and Deokjeok for the haze (February 23–28, 2014) and clean (March 5–9, 2014) periods.**

| | Components | | Seoul | | Deokjeok | |
|---|---|---|---|---|---|---|
| | | | Haze | Clean | Haze | Clean |
| Mass concentrations | PM$_{10}$ (µg m$^{-3}$) | | 143 ± 25 | 39 ± 11 | 100 ± 35 | 28 ± 11 |
| | PM$_{2.5}$ (µg m$^{-3}$) | | 116 ± 29 | 23 ± 10 | 84 ± 31 | 18 ± 9 |
| | | PM$_{2.5}$/PM$_{10}$ | 0.81 ± 0.01 | 0.57 ± 0.10 | 0.84 ± 0.03 | 0.65 ± 0.07 |
| Inorganic species | SO$_4^{2-}$ (µg m$^{-3}$) | | 34.9 ± 9.1 | 3.9 ± 1.4 | 29.2 ± 12.4 | 4.7 ± 2.6 |
| | NO$_3^-$ (µg m$^{-3}$) | | 32.8 ± 8.4 | 4.6 ± 4.2 | 11.4 ± 8.5 | 2.8 ± 3.0 |
| | Cl$^-$ (µg m$^{-3}$) | | 1.1 ± 0.4 | 0.3 ± 0.2 | 0.6 ± 0.6 | 0.2 ± 0.1 |
| | NH$_4^+$ (µg m$^{-3}$) | | 21.6 ± 4.3 | 2.7 ± 1.6 | 14.4 ± 6.0 | 2.4 ± 1.8 |
| | K$^+$ (µg m$^{-3}$) | | 0.9 ± 0.2 | 0.2 ± 0.1 | 0.7 ± 0.3 | 0.2 ± 0.1 |
| | Ca$^{2+}$ (µg m$^{-3}$) | | 0.3 ± 0.1 | 0.1 ± 0.0 | 0.2 ± 0.1 | 0.1 ± 0.0 |
| | Mg$^{2+}$ (µg m$^{-3}$) | | 0.1 ± 0.0 | 0.1 ± 0.0 | 0.1 ± 0.0 | 0.1 ± 0.1 |
| | Na$^+$ (µg m$^{-3}$) | | 0.1 ± 0.0 | 0.1 ± 0.0 | 0.1 ± 0.1 | 0.1 ± 0.1 |
| | | SOR | 0.44 ± 0.04 | 0.15 ± 0.03 | 0.39 ± 0.04 | 0.20 ± 0.06 |
| | | NOR | 0.15 ± 0.02 | 0.05 ± 0.02 | 0.30 ± 0.12 | 0.39 ± 0.19 |
| | | [NH$_4^+$]/[SO$_4^{2-}$] | 1.68 ± 0.16 | 1.74 ± 0.38 | 1.34 ± 0.30 | 1.29 ± 0.25 |
| | SIA (µg m$^{-3}$) | | 89 ± 21 | 11 ± 7 | 55 ± 23 | 10 ± 7 |
| | | SIA/PM$_{2.5}$ | 0.76 ± 0.07 | 0.47 ± 0.09 | 0.65 ± 0.04 | 0.51 ± 0.10 |
| | | SO$_4^{2-}$/PM$_{2.5}$ | 0.30 ± 0.04 | 0.18 ± 0.04 | 0.34 ± 0.05 | 0.26 ± 0.03 |
| | | NO$_3^-$/PM$_{2.5}$ | 0.28 ± 0.03 | 0.18 ± 0.07 | 0.13 ± 0.07 | 0.13 ± 0.07 |
| | | NH$_4^+$/PM$_{2.5}$ | 0.18 ± 0.01 | 0.11 ± 0.02 | 0.17 ± 0.01 | 0.12 ± 0.03 |
| Carbonaceous species | OC (µg m$^{-3}$) | | 14.4 ± 2.5 | 4.9 ± 0.8 | 8.6 ± 2.9 | 2.7 ± 0.6 |
| | EC (µg m$^{-3}$) | | 2.0 ± 0.2 | 1.4 ± 0.2 | 1.2 ± 0.3 | 0.7 ± 0.4 |
| | | OC/EC | 7.3 ± 1.1 | 3.7 ± 1.1 | 7.4 ± 1.7 | 4.9 ± 2.0 |
| Gaseous species | SO$_2$ (ppb) | | 10.6 ± 1.5 | 5.0 ± 0.5 | 11.3 ± 3.4 | 4.5 ± 2.1 |
| | NO$_2$ (ppb) | | 68.1 ± 11.7 | 30.0 ± 10.5 | 11.1 ± 9.5 | 1.1 ± 0.2 |
| | CO (ppm) | | 1.1 ± 0.2 | 0.4 ± 0.1 | 0.7 ± 0.1 | 0.3 ± 0.1 |
| | O$_3$ (ppb) | | 16.8 ± 7.6 | 28.6 ± 6.7 | 57.3 ± 9.5 | 48.6 ± 0.7 |
| Meteorological factors | Temperature (°C) | | 6.2 ± 2.1 | 0.7 ± 1.1 | – | – |
| | Relative humidity (%) | | 55 ± 8 | 49 ± 10 | – | – |
| | Wind speed (m s$^{-1}$) | | 2.3 ± 0.5 | 3.4 ± 0.5 | – | – |
| | Boundary layer height (m) | | 380 ± 60 | 1000 ± 140 | – | – |
| | Daytime (07–18H) visibility (km) | | 3.4 ± 1.1 | 17.1 ± 2.6 | – | – |





**Table 2: The average and standard deviation of analyzed organic compound concentrations and diagnostic ratios in Seoul and Deokjeok for the haze (February 23–28, 2014) and clean (March 5–9, 2014) periods.**

| Organic compounds | Components | Seoul | | Deokjeok | |
|---|---|---|---|---|---|
| | | Haze | Clean | Haze | Clean |
| $n$-Alkanes (ng m$^{-3}$) | | $80 \pm 19$ | $19 \pm 5$ | $34 \pm 12$ | $10 \pm 4$ |
| | CPI$_{odd}$ | $2.0 \pm 0.3$ | $1.7 \pm 0.1$ | $1.7 \pm 0.1$ | $1.7 \pm 0.2$ |
| | Wax C$_n$ (%) | $35 \pm 5$ | $27 \pm 3$ | $28 \pm 4$ | $28 \pm 5$ |
| PAHs (ng m$^{-3}$) | | $18 \pm 12$ | $7 \pm 1$ | $9 \pm 9$ | $3 \pm 1$ |
| | BaP/(BaP + BeP) | $0.34 \pm 0.08$ | $0.37 \pm 0.03$ | $0.28 \pm 0.13$ | $0.32 \pm 0.04$ |
| | IncdP/(IncdP + BghiP) | $0.59 \pm 0.06$ | $0.48 \pm 0.01$ | $0.61 \pm 0.04$ | $0.42 \pm 0.05$ |
| | FLA/(FLA+PYR) | $0.55 \pm 0.01$ | $0.56 \pm 0.01$ | $0.60 \pm 0.02$ | $0.59 \pm 0.00$ |
| | BaA/(BaA + CHR) | $0.26 \pm 0.03$ | $0.28 \pm 0.03$ | $0.22 \pm 0.02$ | $0.22 \pm 0.03$ |
| | ANT/(ANT + PHE) | $0.09 \pm 0.02$ | $0.11 \pm 0.01$ | $0.06 \pm 0.02$ | $0.10 \pm 0.02$ |
| Monocarboxylic acids (ng m$^{-3}$) | | $363 \pm 58$ | $138 \pm 28$ | $166 \pm 49$ | $98 \pm 18$ |
| | $cis$-Pinonic acid (ng m$^{-3}$) | $5.3 \pm 1.7$ | $3.0 \pm 1.0$ | $4.4 \pm 1.6$ | $1.8 \pm 0.2$ |
| | C$_{18:0}$/C$_{18:1}$ | $5.9 \pm 4.5$ | $7.2 \pm 3.1$ | $9.2 \pm 3.0$ | $11.1 \pm 2.9$ |
| Sugars (ng m$^{-3}$) | | $248 \pm 21$ | $148 \pm 28$ | $109 \pm 43$ | $57 \pm 18$ |
| | Levoglucosan (ng m$^{-3}$) | $229 \pm 18$ | $136 \pm 23$ | $98 \pm 39$ | $52 \pm 15$ |
| Dicarboxylic acids (ng m$^{-3}$) | | $522 \pm 137$ | $88 \pm 33$ | $355 \pm 151$ | $80 \pm 47$ |
| | Dicarboxylic acids-C/OC | $0.038 \pm 0.009$ | $0.017 \pm 0.004$ | $0.041 \pm 0.006$ | $0.028 \pm 0.010$ |
| | O:C ratio | $0.90 \pm 0.02$ | $0.73 \pm 0.01$ | $0.92 \pm 0.05$ | $0.81 \pm 0.05$ |



**Table 3: Comparisons of chemical and source characteristics of PM$_{2.5}$ in between Seoul and Deokjeok.**

| Characteristics | Seoul (Downwind urban) | Deokjeok (Upwind background) | Components |
|---|---|---|---|
| Fine mode particles | – | Slightly more fine mode particle fraction | PM$_{2.5}$/PM$_{10}$ |
| Primary aerosols | Higher concentrations (Local emissions + Regional transport) | Lower concentrations (Regional transport) | EC $n$-Alkanes, PAHs, Sugars, Monocarboxylic acids |
| Secondary inorganic aerosols (SIA) | Larger proportion of PM$_{2.5}$ during the haze period due to higher NO$_3^-$/PM$_{2.5}$ in Seoul (Local NO$_x$ emissions) | Larger proportion of PM$_{2.5}$ during the clean period due to higher SO$_4^{2-}$/PM$_{2.5}$ in Deokjeok (Regional transport of SO$_2$ and SO$_4^{2-}$) | SIA/PM$_{2.5}$ NO$_3^-$/PM$_{2.5}$ SO$_4^{2-}$/PM$_{2.5}$ |
| Secondary organic aerosols (SOA) | Smaller fraction | Larger fraction | OC/EC Dicarboxylic acids-C/OC |
| Aging, oxidation, and photochemistry | Fresher | More aged | BaP/(BaP + BeP) |
| | Less oxidation | More oxidation | Dicarboxylic acids O:C SOR |
| | Shorter residence time | Longer residence time | C$_{18:0}$/C$_{18:1}$ |
| Source characteristics | More biogenic VOCs and precursors (Urban forest) | Less biogenic VOCs and precursors (Marine island) | CPI$_{odd}$ Wax C$_n$ $cis$-Pinonic acid |
| | More biomass burning (Local and regional sources) | Less biomass burning (Regional source) | Levoglucosan |
| | Grass, wood, and coal combustion sources but more fossil fuel combustion-like | Grass, wood, and coal combustion sources | FLA/(FLA + PYR) |
| | Coal combustion sources, but more vehicular emission-like | Coal combustion sources, but more petrogenic-like | BaA/(BaA + CHR) |
| | More pyrogenic-like | Petrogenic (Haze period) | ANT/(ANT + PHE) |
| | Petroleum combustion, but more biomass and coal combustion-like (Clean period) | Petroleum combustion, but more petrogenic-like (Clean period) | IncdP/(IncdP + BghiP) |



**Table 4: Comparisons of meteorological conditions and properties of PM$_{2.5}$ during between the haze and clean periods.**

| Characteristics | Haze period (February 23–28) | Clean period (March 5–9) | Components |
|---|---|---|---|
| Meteorological conditions | Warm and humid<br>More stagnant (Low winds)<br>Low boundary layer height | Cold and dry<br>Less stagnant (High winds)<br>High boundary layer height | Meteorological factors in Seoul |
| Fine mode particles | Dominant (> 80% of PM$_{10}$ in Seoul) | Less dominant (< 60% of PM$_{10}$ in Seoul) | PM$_{2.5}$/PM$_{10}$ |
| Primary aerosols | Higher concentrations<br>(Accumulation in the shallow boundary layer) | Lower concentrations<br>(Ventilation by the high winds) | EC<br>$n$-Alkanes, PAHs,<br>Monocarboxylic acids,<br>Sugars |
| Secondary inorganic aerosols (SIA) | Larger proportion of PM$_{2.5}$<br>(~ 76% of PM$_{2.5}$ in Seoul) | Smaller proportion of PM$_{2.5}$<br>(~ 47% of PM$_{2.5}$ in Seoul) | SIA/PM$_{2.5}$<br>NO$_3^-$/PM$_{2.5}$<br>SO$_4^{2-}$/PM$_{2.5}$ |
| Secondary organic aerosols (SOA) | Larger fraction<br>(OC/EC ~ 7.3 in Seoul) | Smaller fraction<br>(OC/EC ~ 3.7 in Seoul) | OC/EC<br>Dicarboxylic acids-C/OC |
| Aging, oxidation, and photochemistry | More aged | Fresher | BaP/(BaP + BeP) |
| | More oxidation<br>(Warm and stagnant condition) | Less oxidation<br>(Cold and ventilation effect) | Dicarboxylic acids O:C<br>SOR |
| | Enhanced photochemistry | Weakened photochemistry | $cis$-Pinonic acid |
| Source characteristics | Grass, wood, and coal combustion sources | Petroleum combustion sources | IncdP/(IncdP + BghiP) |
| | Petrogenic (Deokjeok) | Pyrogenic-like | ANT/(ANT + PHE) |





**Table 5: Correlation coefficients between meteorological variables and each component in Seoul for the measurement period of February 23–March 9, 2014.**

| | Temperature | Relative humidity | Wind speed | Boundary layer height |
|---|---|---|---|---|
| PM$_{2.5}$ | +0.68 ** | +0.49 | −0.70 ** | −0.89 ** |
| SO$_4^{2-}$ | +0.64 * | +0.47 | −0.61 ** | −0.82 ** |
| NO$_3^-$ | +0.68 ** | +0.53 * | −0.71 ** | −0.87 ** |
| NH$_4^+$ | +0.66 ** | +0.50 | −0.67 ** | −0.86 ** |
| OC | +0.65 ** | +0.33 | −0.79 ** | −0.89 ** |
| EC | +0.57 * | +0.03 | −0.66 ** | −0.65 ** |
| $n$-alkanes [†] | +0.81 ** | +0.38 | −0.79 ** | −0.89 ** |
| PAHs [†] | +0.32 | +0.33 | −0.45 | −0.50 |
| Monocarboxylic acids [†] | +0.81 ** | +0.25 | −0.87 ** | −0.92 ** |
| Sugars [†] | +0.62 * | +0.26 | −0.71 ** | −0.81 ** |
| Dicarboxylic acids [†] | +0.62 * | +0.53 | −0.63 * | −0.83 ** |
| SOR | +0.72 ** | +0.59 * | −0.64 * | −0.88 ** |
| NOR | +0.69 ** | +0.67 ** | −0.60 * | −0.87 ** |
| OC/EC ratio | +0.65 ** | +0.51 * | −0.72 ** | −0.90 ** |
| Dicarboxylic acids O:C ratio [†] | +0.71 ** | +0.54 * | −0.60 * | −0.87 ** |

[†] Data for organic compounds on February 25 are not available.

[*] Statistically significant at 95% confidence level.

[**] Statistically significant at 99% confidence level.



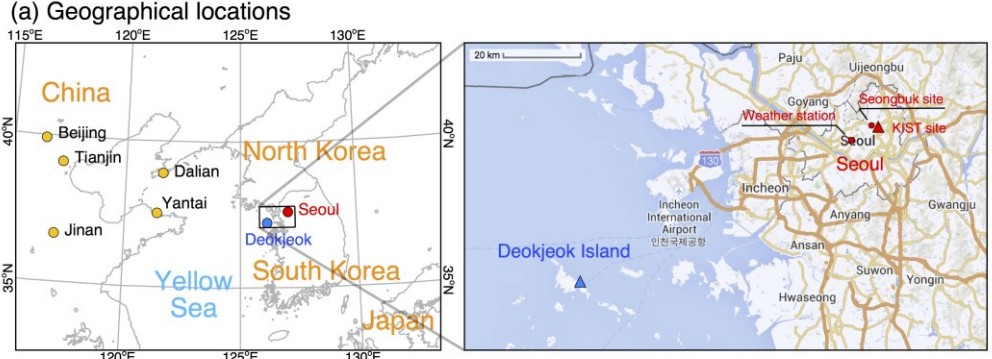

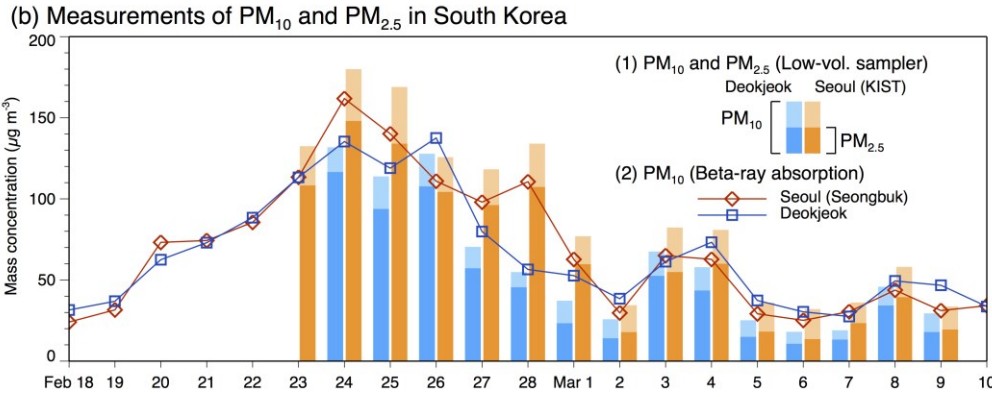

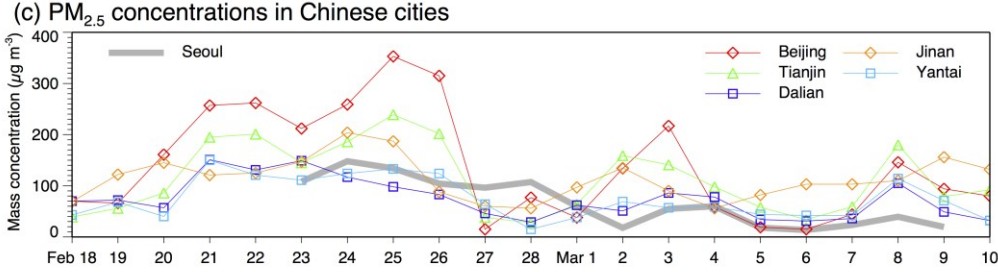

**Figure 1: (a)** Geographical locations of filter-sampling sites (KIST site in Seoul and Deokjeok site), KMOE air quality monitoring sites (Seongbuk site in Seoul and Deokjeok site), and KMA weather station in Seoul. **(b)** Daily mass concentrations of PM$_{10}$ and PM$_{2.5}$ sampled at the KIST site in Seoul and Deokjeok site (bars) during the multi-day haze episode and following the clean period in late February to early March of 2014. Red diamonds and blue squares with solid lines denote 24-h averages of PM$_{10}$ concentration measured by beta-ray absorption at the Seongbuk site in Seoul and at the Deokjeok site, respectively. **(c)** Daily PM$_{2.5}$ concentrations in five Chinese cities of Beijing, Tianjin, Dalian, Jinan, and Yantai (http://www.tianqihoubao.com/aqi/).



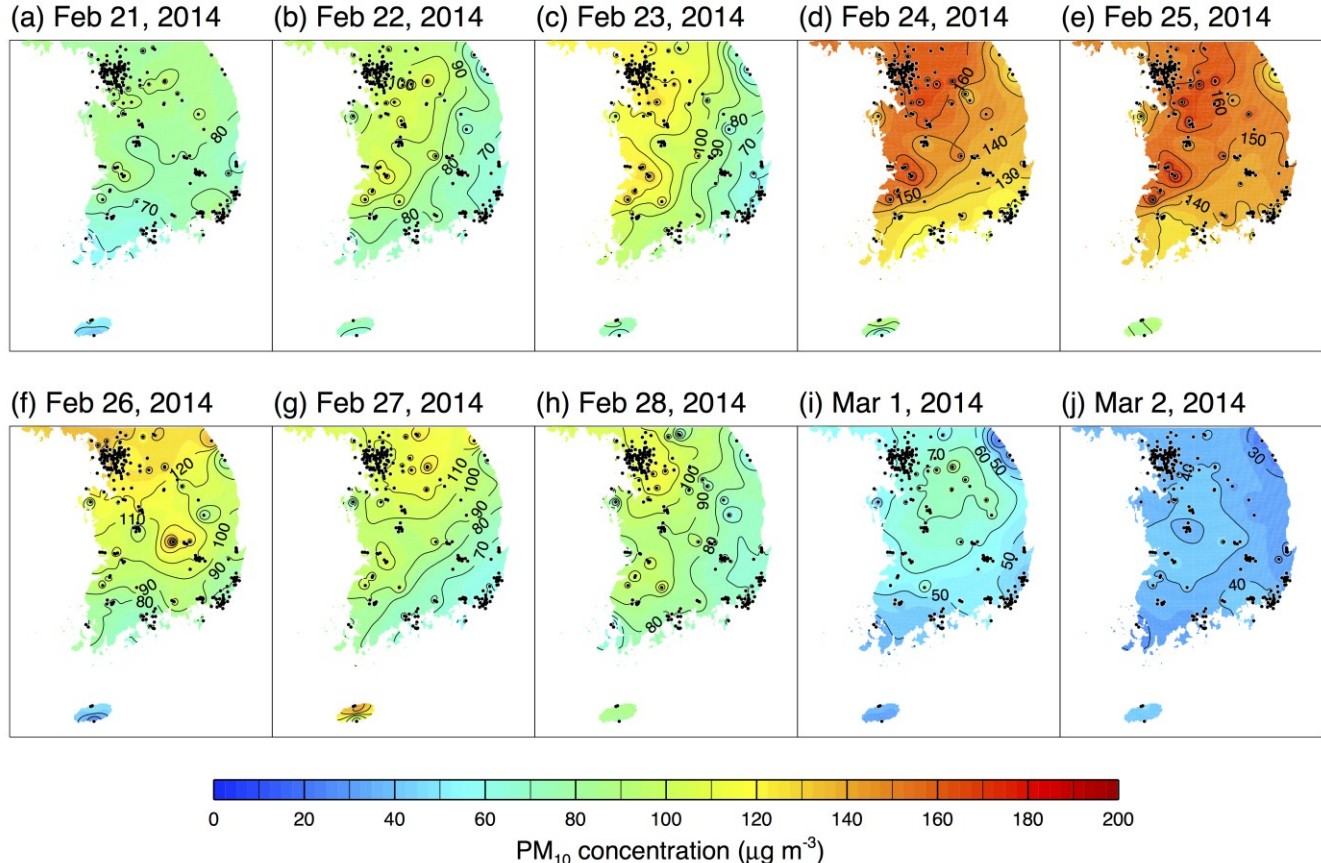

**Figure 2: Spatial distribution of PM$_{10}$ concentration for the period of February 21–March 2, 2014 using data from 247 air quality monitoring sites (black dots) in the South Korea.**

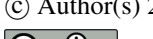



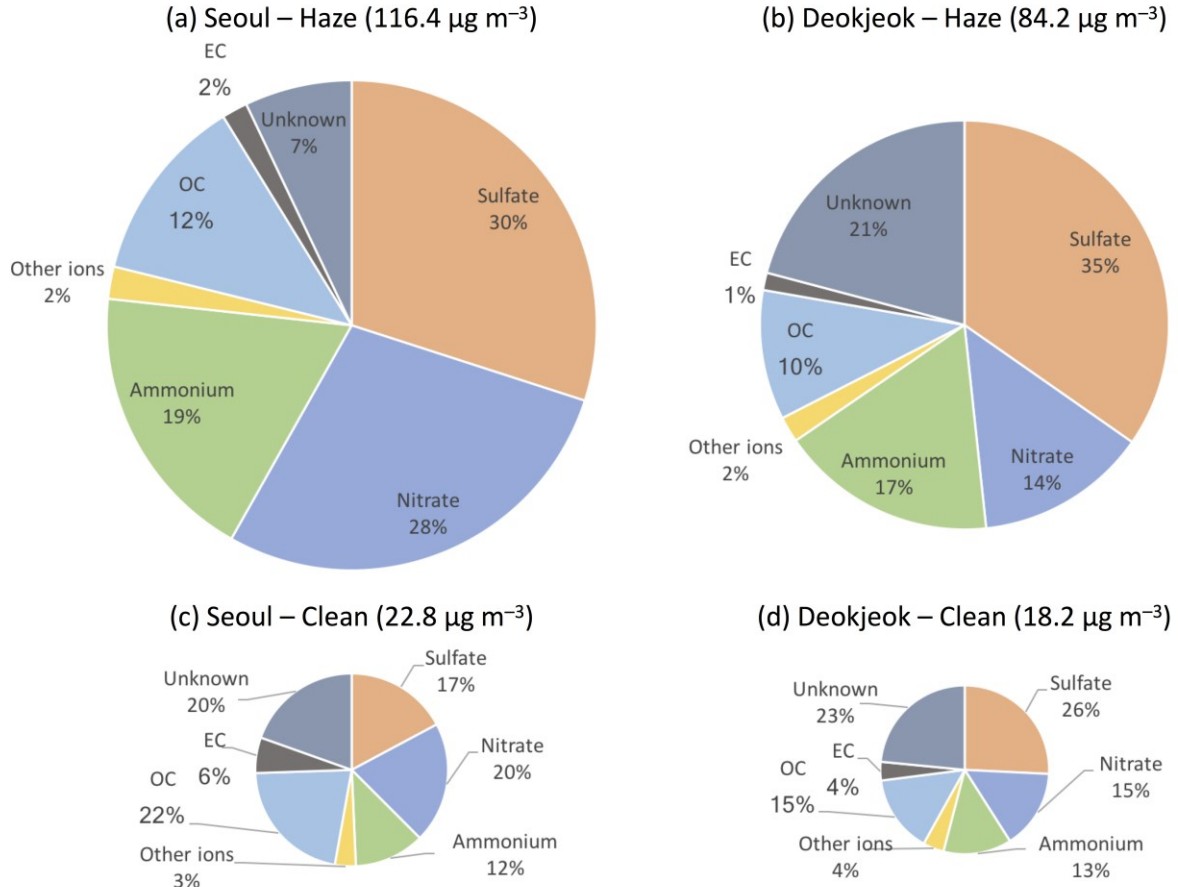

**Figure 3: Mass fraction (%) of each component to the total PM$_{2.5}$ mass in Seoul and Deokjeok during the haze and clean periods. The size of each chart is proportional to the averaged PM$_{2.5}$ mass concentration.**



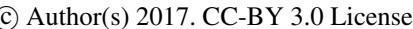

**Figure 4: Daily time series of gas-phase pollutants (SO₂, NO₂, CO), secondary inorganic aerosol compounds (nss-SO₄²⁻ and NO₃⁻) and SOR, carbonaceous species (EC and OC), and meteorological factors (temperature, specific humidity, wind speed, and boundary layer height) during the analysis period.**





**Figure 5: Daily time-series of (a) *n*-alkanes, (b) polycyclic aromatic hydrocarbons (PAHs), (c) monocarboxylic acids, (d) sugars, (e) dicarboxylic acids, and (f) O:C ratio of the dicarboxylic acids measured at KIST site in Seoul and Deokjeok site for the analysis period.**





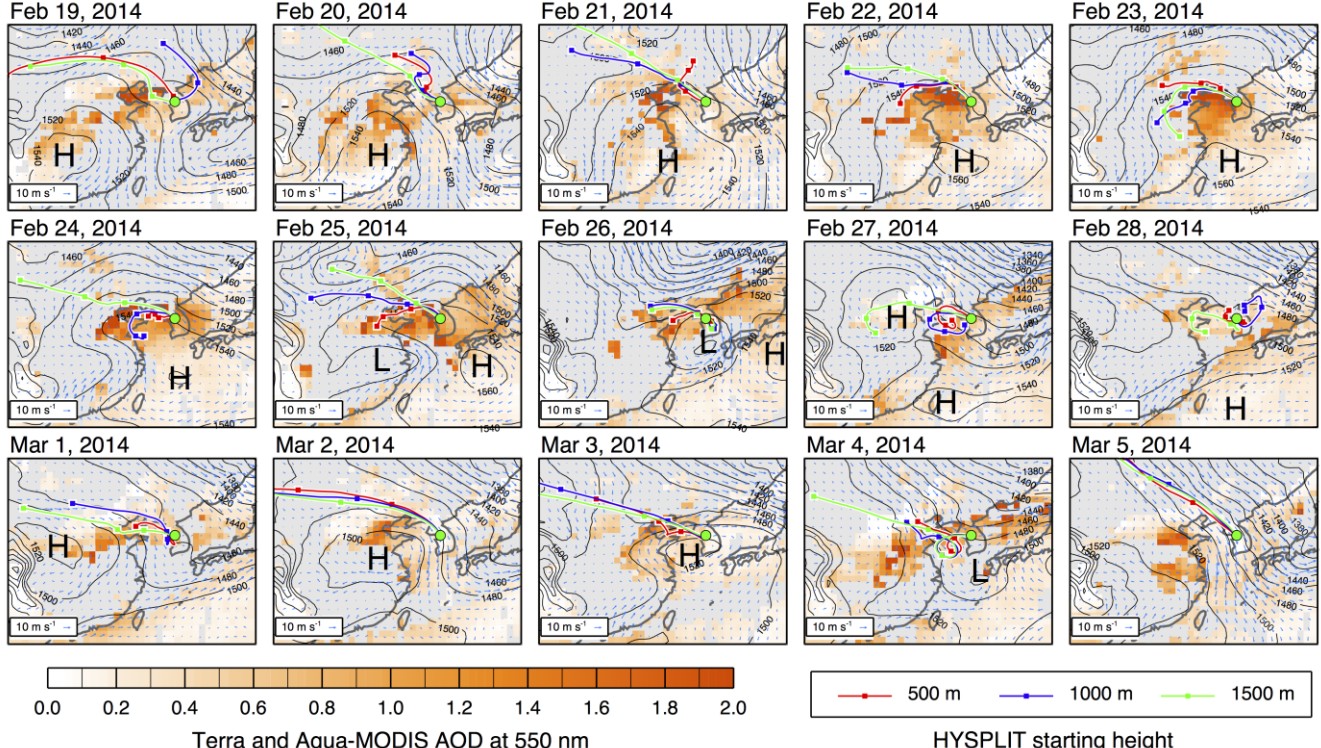

**Figure 6: Daily evolution of synoptic meteorological conditions over East Asia during the multi-day haze episode in 2014. ERA-Interim daily mean geopotential height (contours, unit of gpm) and wind (arrows) at 850 hPa are superimposed on the Terra and Aqua-MODIS AOD at 550 nm. Red, blue, and green lines with squared respectively represent backward trajectories from 500 m, 1000 m, and 1500 m above sea level over the sampling site at 21:00 local time (GMT+0900) on each day with endpoints of 24-h interval.**





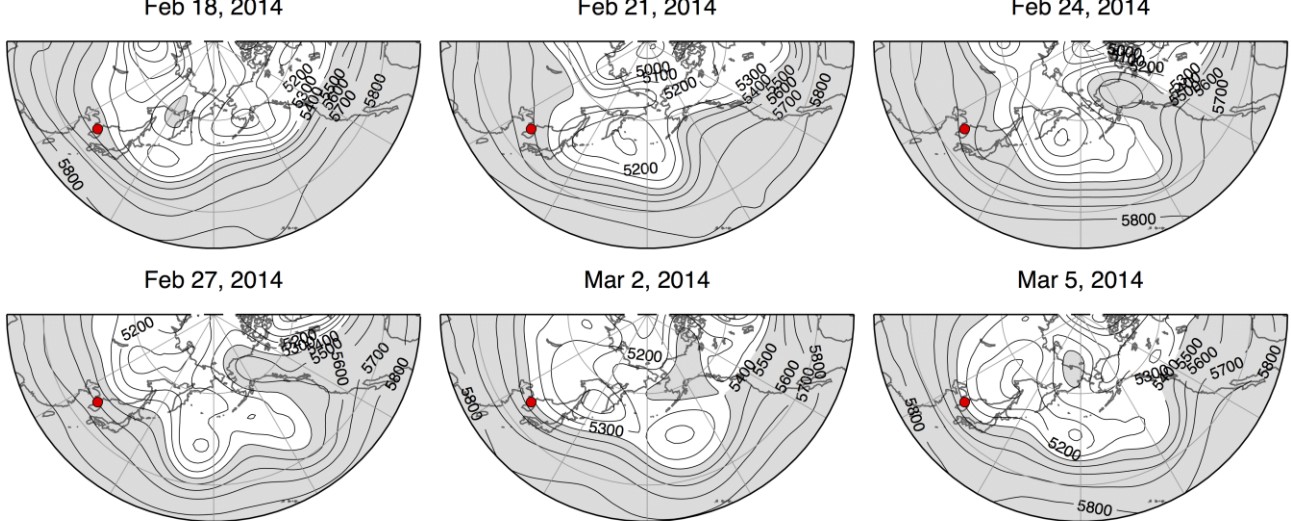

**Figure 7: ERA-Interim daily mean geopotential height (contours, unit of gpm) at 500 hPa. Seoul is marked as red filled circles, and the areas of which geopotential height is higher than 5400 gpm are shaded.**