# Peer review of "On the multi-day haze in the Asian continental outflow: An important role of synoptic condition combined with regional and local sources"

_Atmospheric Chemistry and Physics, 2016_

## Referee Comment (RC1) · Anonymous Referee #1 · 3 May 2017

The paper by Seo et al. summarizes observations at a background and an urban site (Deokjeok Island and Seoul) during a multi-day haze episode in east Asia. The measurements are focused on aerosol composition and auxiliary gases. Meteorological conditions and weather systems associated with this event are also discussed. The main conclusion of the paper is that in Seoul, both regional transport and local emissions contribute to haze formation when meteorological conditions are favorable. The paper is overall well organized and well written. I support publishing the paper after the following minor comments are addressed:

1. p. 3, L 16: PM10 was also measured at both sites, correct? 2. p. 4, L7: what precautions were taken to reduce gas-phase artifacts in the OC filter measurements?

any back-up filters used? If not, please comment on the possible effects on the re-ported results. 3. p. 4, L 21: what was the total volume of the solvent mixture? Also, rephrase as "....and methanol (3:1; v/v) twice, each for 30 min." 4. p. 4, L23: Please comment on the possible evaporation artifacts when the samples are heated to 40 C. 5. p. 7, L 10: Considering the distance between Deokjeok Island and Seoul and the wind speed of 2 m/s during the haze period, it seems there would be a 12 hr transport time for plumes to travel from Deokjeok Island to Seoul. Couldn't secondary formation of aerosols during this time also contribute to higher pollution levels in Seoul compared to Deokjeok? 6. p. 7, L26: I don't think comparing CO and PM2.5 is correct since CO is a primary pollutant and PM2.5 is predominantly a secondary pollutant. 7. p.7, L30: PM2.5/PM10 values are not significantly different at the two sites, so I would remove the 2nd sentence in section 3.2.3. 8. p.8, L7: Contribution of HNO3 to NOR is not con-sidered (possibly because the measurements were not available). Since temperature and RH affect partitioning of gas phase nitric acid to aerosols, without HNO3, NOR as defined is not that useful. I suggest removing NOR from the discussions in the text and tables. 9. p.8, L10: SIA fraction in Deokjeok is 51%, not 57%. 10. p.8, section 3.2.3 and 3.3: Although it's true that Seoul measurements include contributions from local emissions, I think it's also important to indicate that differences in boundary layer heights in Seoul vs. Deokjeok can also contribute to some of the observed differences in Fig. 4-5. 11. p. 9, L4: what are petrogenic sources specifically and how are they dif-ferent than petroleum combustion processes? 12. p.9, the paragraph on the fractions of PAHs and emission sources is lengthy and at the end, it seems each pair of PAHs suggest one type of source impact. I suggest rephrasing and shortening this section. 13. p. 9, L22: As somebody who's not familiar with seasons in Seoul, it's surpris-ing that measurements in Feb. would indicate biogenic alkane emissions from plant waxes. Aren't trees dormant in Feb in Seoul? Also related to the biogenic alkanes... it seems only C20-C36 alkanes that are found in biogenic emissions are characterized here. This certainly skews the results since midsize semivolatile alkanes are mostly as-sociated with vehicular emissions (see e.g. Gentner et al., EST, 2016). This needs to

be addressed. 14. p. 10, L4: which carboxylic acid can originate as a primary aerosol component from fossil fuel combustion? The low O/C content of POA in many urban environments suggests primary OA does not contain such oxygenated compounds like carboxylic acids. This is again repeated in p. 12, L3-5. Higher acidic components in Seoul would suggest local SOA production rather than contribution from primary emissions. 15. p. 11, L18: To fully understand aerosol nitrate formation, some exercise with a thermodynamic model is needed because equilibrium partitioning of $HNO_3$ to $NO_3^-$ is RH and acidity dependent. Therefore, the high correlation of $NO_3^-$ with RH doesn't necessarily mean $NO_3^-$ was produced aqueously by uptake of $N_2O_5$. If the authors mean uptake of $HNO_3$ instead (L20) then the process is not aqueous chemistry, but rather shifts in equilibrium partitioning and dissolution of $HNO_3$. 16. p. 11, L26: it was indicated on L15 (P11) that $NO_2$ decreased during 2/27-3/2 with high wind speeds, but that's not similar to the observations in CO that showed little difference with wind speed. Please clarify. 17. p. 11, L31-33: It's unclear why trends in organic tracers suggests regional transport of $SO_2$. 18. p. 12, L7: Why is the O:C of carboxylic acids only explored? The conclusions drawn are acceptable if O:C represented values for all components of OA. 19. p. 12, L12-13: I disagree with the conclusion here. If RH and OC are not well correlated while RH and OC/EC is, this suggests to me that EC and RH are somehow correlated, but still EC is solely a primary tracer, I think the correlation is merely due to meteorology and cannot suggest anything about aq-phase production of OC.

Figures Fig. 2- please add the sampling site locations to one of the panels. Fig. 3- I understand that OC was the parameter directly measured, but since the pie charts represent total aerosol mass from each species, why not convert OC to OM, using appropriate, representative ratios of OM/OC? Fig. 6- I suggest marking the political boundaries with a different colors than the contours. Also, the wind arrows don't show up well. Consider making the panels larger. It would also help if the locations of the sampling sites are marked on one panel.

---

## Referee Comment (RC2) · Anonymous Referee #2 · 10 May 2017

MS No.: acp-2016-1184

"On the multi-day haze in the Asian continental outflow: An important role of synoptic condition combined with regional and local sources" by Seo et al.

This paper discusses the results of a study of PM2.5 bound inorganic - organic species and gaseous pollutants on the haze events in February 2014 at the outflow regions of Asian pollution. Meteorological factors and synoptic conditions are also discussed along with chemistry for the same context. These data of Asian outflow were collected at a site in highly industrialized region of Seoul and at a background site, Deokjeok Island, over the Yellow Sea. Paper contents important data and interesting discussions on the topic, hence I recommend for the publication in ACP. However, I have

several comments/suggestions, especially at several places where statements are contradictory which should be addressed before making a final decision. P1,L16-18: Not clear. It is an overall general statement for both the sites. Better to summarise according to individual site, as in the following sentences. P4,L3-10: Do authors think that it is logically correct to measure PM2.5 mass using low-volume sampler and compare PM2.5 chemical measurement using high-volume sampler. Possible biases should be properly addressed here with references (e.g. MAPAN, 2013, Volume 28, Issue 3, pp 153–166). P4,L12-13: The readability/ sensitivity of microbalance should be mentioned here. P6,L13-16: Back trajectory analysis should also be discussed in support. P6,L17-18: Not clear. P6,L33: "compared and characterized in Table 4" should be "compared and summarised in Table 4" P7,L3: Please mention (number) boundary layer height and wind speed here. P7,L7-10: Backward trajectory should also be discussed here. P7,L12-18: Section 3.2.2, this is in contrast to section 3.2.1 and following sections, e.g. P13,L10-12, and several other places (please see comments below). P8,L1-2: How about the contribution of marine aerosols, especially at Deokjeok? Mass concentration of PM2.5 and PM10 should also consider while discussing PM2.5/PM10 ratios. P8,L15: For more clarity, it is better to also use nssSO42- concentration in discussions. P8L20-21: OC/EC ratios in Deokjeok and Seoul are 7.4±1.7 and 7.3±1.1, respectively in haze period. Do authors think "The OC/EC ratio is higher in Deokjeok than in Seoul during both haze and clean periods." is a correct sentence if authors see the values in view of statistical significance? Similarly check such statements in other places as well. P9,L4-16: Discussion is contradictory in context of sources discussed before this para and later in the text. P9,L18: "total sugar" should be "total sugar identified" P9,L31: Reference is needed. P9L33; P10,L1-2: Again contradictory statements (as pointed out above). P10,L10: "OC/EC ratio" same comment as in P8L20-21. P10, Section 3.3.2: Result suggests that particles are fresh in Seoul and comparatively aged in Deokjeok. An analysis of fresh and aged PM2.5, PM10 should be incorporated. P11, L5-8: How about boundary layer height? P11,L10-22: I suggest to check and discuss the relation of RH, NO2 with

sulfate formation apart from photochemical formation. P12,L1-2: I suggest to check and discuss primary and secondary OC contribution in haze and clean days at both the site to justify this statement. P13,L10-12: Not clear.

Please also note the supplement to this comment:
http://www.atmos-chem-phys-discuss.net/acp-2016-1184/acp-2016-1184-RC2-supplement.pdf

---

## Author Comment (AC1) · 20 Jun 2017

**Response to Anonymous Referee #1**

The paper by Seo et al. summarizes observations at a background and an urban site (Deokjeok Island and Seoul) during a multi-day haze episode in east Asia. The measurements are focused on aerosol composition and auxiliary gases. Meteorological conditions and weather systems associated with this event are also discussed. The main conclusion of the paper is that in Seoul, both regional transport and local emissions contribute to haze formation when meteorological conditions are favorable. The paper is overall well organized and well written. I support publishing the paper after the following minor comments are addressed:

We appreciate the reviewer for careful reading and helpful comments that improve quality of the manuscript. As indicated in the following point-by-point responses, we have incorporated the reviewer's comments and suggestions into the revised manuscript. We have conducted additional analyses, modified texts, figures, and tables, and added several new figures (Figs. 2 and 5) and references in the revised manuscript. Each response to reviewer colored in blue and changes in the manuscript colored in red.

1. p. 3, L 16: $PM_{10}$ was also measured at both sites, correct?

Yes, it was. We used measured $PM_{10}$ mass concentrations for (i) comparisons with the $PM_{10}$ data from the air quality monitoring network (beta attenuation monitoring) and (ii) comparisons of $PM_{2.5}/PM_{10}$ ratios between haze and clean periods. We replaced "$PM_{2.5}$" in L16 on p.3 with "PM" in the revised manuscript.

2. p. 4, L7: what precautions were taken to reduce gas-phase artifacts in the OC filter measurements? any back-up filters used? If not, please comment on the possible effects on the reported results.

In this study, we did not use both organic denuder and backup filters in the high-volume air sampler, which help to correct positive and negative OC artifacts. In addition, as the other reviewer pointed out, using the different-volume air samplers for inorganic and carbonaceous species could result in biases in estimation of particle concentrations. Considering these two factors, we added a new paragraph that addresses probable artifacts resulted from the measurement at the end of Sect. 2.1, as follows:

Note that the $PM_{2.5}$ sampling conducted in this study could result in artifacts of carbonaceous species. Firstly, we used the high-volume air sampler for carbon analyses, while the total mass and ion concentrations were obtained by the low-volume air sampler. Secondly, we did not employ preceding organic denuder and backup filters in the high-volume air sampler for correction of both positive artifacts (by adsorption of organic vapor) and negative artifacts (by volatilization of semivolatile materials) of measured OC. Although the positive artifacts are thought to be larger than the negative artifacts (Chow et al., 2010; Kim et al., 2016), the high-volume air sampler tends to underestimate particle concentrations (Lagler et al., 2011; Aggarwal et al., 2013) and thus the measurement by the high-volume air sampler may partly reduce such positive artifacts as recently estimated by Kim et al. (2016). However, aging and oxidation properties and source characteristics of measured aerosols can still be altered by potential loss of the semivolatile organic compounds. In this study, therefore, we used organic compounds more for qualitative comparisons between different places and periods rather than for quantitative analysis.

3. p. 4, L 21: what was the total volume of the solvent mixture? Also, rephrase as "....and methanol (3:1; v/v) twice, each for 30 min."

The total volume of the solvent mixture was 100 mL because we conducted ultrasonication of the one filter sample for two times in each of two 50 mL solvent mixture. To clarify the procedure, we modified the sentence as follows:

To identify and measure concentrations of individual organic compounds, one-half of the quartz fiber filter was used and ultrasonicated twice in each of two 50 mL mixtures of dichloromethane and methanol (3:1; v/v) in sequential order, each for 30 min (total volume of the solution: 100 mL).

4. p. 4, L23: Please comment on the possible evaporation artifacts when the samples are heated to 40 C.

We added a sentence to address the unavailability of light compounds that resulted from the probable evaporation in the analytic process, as follows:

This process induces low recovery rates (~70%) of relatively low molecular weight compounds (e.g. $n$-alkanes lighter than $C_{20}$ and PAHs lighter than anthracene), and we excluded these compounds from further analyses.

5. p. 7, L 10: Considering the distance between Deokjeok Island and Seoul and the wind speed of 2 m/s during the haze period, it seems there would be a 12 hr transport time for plumes to travel from Deokjeok Island to Seoul. Couldn't secondary formation of aerosols during this time also contribute to higher pollution levels in Seoul compared to Deokjeok?

The "local contribution" that we mentioned in this sentence intrinsically includes both contributions by (i) the local primary emissions and (ii) the secondary formation "at local" from both transported and locally emitted precursor gases. To clarify this, we modified the sentence as follows:

On the other hand, the tendency of the lower mass concentration of each aerosol component in Deokjeok than that in Seoul (Table 1) implies the local contributions of both primary emissions from the SMA and secondary productions from local and transported precursor gases to the long-lasting haze in Seoul.

6. p. 7, L26: I don't think comparing CO and $PM_{2.5}$ is correct since CO is a primary pollutant and $PM_{2.5}$ is predominantly a secondary pollutant.

We agreed with the reviewer's comment and removed "similar characteristics to total $PM_{2.5}$;" from the sentences, as follows:

CO, which is an incomplete combustion product of fossil fuels or biomass/biofuels, is affected by both local emissions in the SMA and the regional transport from China and thus shows higher concentration in Seoul than in Deokjeok and also in the haze period than in the clean period (Table 1).

7. p.7, L30: $PM_{2.5}/PM_{10}$ values are not significantly different at the two sites, so I would remove the 2nd sentence in section 3.2.3.

We agree with the reviewer's comment. Instead of removing this sentence, we modified it to describe significantly high $PM_{2.5}/PM_{10}$ values during the haze period, as follows:

The larger $PM_{2.5}/PM_{10}$ during the haze compared to the clean period in both places (Table 1) suggests more influence of the secondary aerosol formation during the haze period (Irei et al., 2015).

We also deleted the second row (fine mode particles related to $PM_{2.5}/PM_{10}$) from Table 3.

8. p.8, L7: Contribution of $HNO_3$ to NOR is not considered (possibly because the measurements were not available). Since temperature and RH affect partitioning of gas phase nitric acid to aerosols, without $HNO_3$, NOR as defined is not that useful. I suggest removing NOR from the discussions in the text and tables.

We agree with the reviewer's point that contribution of HNO₃ should be considered when we discuss the oxidation of NO$_x$. We removed every words related to "NOR," following the reviewer's suggestion. L7 on p.8 was now changed as follows:

The sulfur oxidation ratio (SOR = $n$ SO$_4^{2-}$ / [$n$ SO$_4^{2-}$ + $n$ SO$_2$]) ($n$ refers to the molar concentration) represents the atmospheric conversion of SO$_2$ to sulfate aerosol through the oxidation and partitioning (Squizzato et al., 2013).

Also, the "NOR" related rows in Tables 1 and 5, as well as related descriptions in the text were deleted in the revised manuscript.

9. p.8, L10: SIA fraction in Deokjeok is 51%, not 57%.

Thanks for the correction. It was now corrected.

10. p.8, section 3.2.3 and 3.3: Although it's true that Seoul measurements include contributions from local emissions, I think it's also important to indicate that differences in boundary layer heights in Seoul vs. Deokjeok can also contribute to some of the observed differences in Fig. 4-5.

The higher concentrations of pollutants in Seoul during the haze could be contributed not only by the more local emissions and secondary formation but also by the shallower boundary layer, if the boundary layer height (BLH) in Seoul was lower than BLH in Deokjeok. However, as shown by Fig. S1, the BLH in Seoul was rather higher than that in Deokjeok during the haze, and thus the contribution of different boundary layer condition in Seoul and Deokjeok to the observed differences in Figs. 4 and 5 (Figs. 6–7 in the revised version) seems to be limited. We added average BLH for haze and clean periods in Deokjeok into Table 1 for additional information.

[Figure]

**Figure S1: Boundary layer height (BLH) from ERA-Interim reanalysis data. (a) Time series of BLH near Seoul (37.5°N, 127.0°E) and Deokjeok (37.0°N, 126.0°E). (b and c) Spatial distribution of BLH averaged for (b) haze period (February 23–28) and (c) clean period (March 5–9). Filled circles and plus symbols indicate locations of the measurement site and nearest ERA-interim grid points from the sites, respectively.**

11. p. 9, L4: what are petrogenic sources specifically and how are they different than petroleum combustion processes?

Petrogenic sources of PAHs are related to spilled or leaked petroleum products (crude oil, fuels, lubricants, etc.), while petroleum combustion is one of the pyrogenic sources of PAHs (fossil fuels combustions or biomass burning) (Stogiannidis and Laane, 2015).

12. p.9, the paragraph on the fractions of PAHs and emission sources is lengthy and at the end, it seems each pair of PAHs suggest one type of source impact. I suggest rephrasing and shortening this section.

To shorten the second paragraph of Sect. 3.3.1, we created a new figure set, Fig. 5 to summarize the PAH ratios from the present study and previously reported PAHs diagnostic ratios with references. Together with Fig. 5, the lengthy paragraph was shortened as follows:

PAHs are combustion byproducts of all types of organic matters, especially related to the incomplete combustion of fossil fuels and biomass burning (Nisbet and LaGoy, 1992; Bi et al., 2003). Various diagnostic ratios with individual PAH species are helpful to search their emission sources (Tobiszewski and Namieśnik, 2012). The average PAH ratios in Seoul and Deokjeok for the haze and clean periods are summarized in Fig. 5 with previously reported diagnostic ratios (Yunker et al., 2002; Pies et al., 2008; De La Torre-Roche et al., 2009; Akyüz and Çabuk, 2010; Oliveira et al., 2011). Interpretation of these PAH ratios all together seems ambiguous and contradictory due to mixing of various emission sources. However, relative comparisons of each ratio between different places and different periods reveal more pyrogenic sources such as fossil fuel combustion and vehicular emissions in Seoul than in Deokjeok (Figs. 5a–c), in the overall influence of coal combustion and/or biomass burning during the haze period (Figs. 5a–b, and 5d).

[Figure]

Figure 5: Various PAH ratios of (a) FLA/(FLA + PYR), (b) BaA/(BaA + CHR), (c) ANT/(ANT + PHE), (d) IcdP/(IcdP + BghiP), and (e) BaP/(BaP + BeP) in Seoul (red diamonds) and Deokjeok (blud squares) during the haze (filled symbols) and clean (opened symbols) periods. Horizontal bars indicate standard deviation.

13. p. 9, L22: As somebody who's not familiar with seasons in Seoul, it's surprising that measurements in Feb. would indicate biogenic alkane emissions from plant waxes. Aren't trees dormant in Feb in Seoul? Also related to the biogenic alkanes... it seems only $C_{20}$-$C_{36}$ alkanes that are found in biogenic emissions are characterized here. This certainly skews the results since midsize semivolatile alkanes are mostly associated with vehicular emissions (see e.g. Gentner et al., EST, 2016). This needs to be addressed.

In South Korea, coniferous forests are about 40% of the total forest area (Lee et al., 2017). Since conifers are not dormant during the cold season, epicuticular wax-related alkanes could be emitted from conifer needles by sloughing and wind abrasion even in winter. In the previous studies on $PM_{10}$ organic compounds in Seoul, concentrations of typical biogenic $n$-alkanes ($C_{29}$, $C_{31}$, and $C_{33}$) in winter was comparable to those in other seasons (see Table 1 in Choi et al., 2016).

In terms of the second point, we agree with the reviewer. Sampling and analytic artifacts that commented previously by the reviewer actually resulted in the low recovery rate of light alkanes, and we excluded these compounds (which may associate with vehicular emissions) from the present study.

Thus, we modified and extended the second sentence of the paragraph (L23–24 on p.9) as follows:

Although the measurements in this study were conducted in the cold season, conifers that occupy ~40% of forest in South Korea (Lee et al., 2017) can be a biogenic source of organic compounds. Whereas short-chain (light) $n$-alkanes are mostly associated with incomplete combustion or vehicle exhaust (Gentner et al., 2017), biosynthetic processes result in long-chain high molecular weight ($C_{27}$–$C_{33}$) $n$-alkanes with distinctive odd-to-even carbon number preference (Simoneit, 1991; Rogge et al., 1993). Note that $n$-alkanes in this study reflect less anthropogenic sources because we analyzed only high molecular weight $n$-alkanes ($C_{20}$–$C_{36}$). The carbon preference index ($CPI_{odd}$) defined by a concentration ratio of odd-to-even numbered $n$-alkane homologues is higher than 3 for more biogenic sources while that is close to 1 for more anthropogenic combustion sources (Simoneit, 1989).

14. p. 10, L4: which carboxylic acid can originate as a primary aerosol component from fossil fuel combustion? The low O/C content of POA in many urban environments suggests primary OA does not contain such oxygenated compounds like carboxylic acids. This is again repeated in p. 12, L3-5. Higher acidic components in Seoul would suggest local SOA production rather than contribution from primary emissions.

Although the major source of dicarboxylic acids is believed to be secondary production (Kundu et al., 2010), their primary sources also have been reported by previous studies: motor exhaust (Kawamura and Kaplan, 1987; Grosjean, 1989), wood combustion (Rogge et al., 1998; Oros and Simoneit, 2001), forest biomass burning (Narukawa et al., 1999; Falkovich et al., 2005), and meat cooking (Rogge et al., 1991). For example, Kawamura and Kaplan (1987) reported that the major dicarboxylic acid compounds in vehicular emission except the most abundant one, oxalic acid, were maleic and phthalic acids, which were also measured in this study. Since L4–6 on p.10 is a general description, we didn't remove the text related to primary source of dicarboxylic acids but slightly modified it with additional references, as follows:

Dicarboxylic acids could originate from the primary sources like fossil fuel combustion or biomass burning (Kawamura and Kaplan, 1987; Rogge et al., 1998), but more from the secondary sources like gas-particle partitioning of semivolatile products from the photooxidation of anthropogenic or biogenic precursors and aqueous chemistry in aerosol waters (Rogge et al., 1993; Kundu et al., 2010; Zhang et al., 2010; Zhang et al., 2016).

Also, L3–5 on p.12 is modified as follows:

As briefly mentioned in Sect. 3.3, dicarboxylic acids mainly originate from the secondary sources. Concentration differences for dicarboxylic acids in Seoul and Deokjeok are small during the clean period, but large during the haze period (except on February 26) probably due to influences of local precursor emissions and secondary formation in Seoul (Fig. 7f).

15. p. 11, L18: To fully understand aerosol nitrate formation, some exercise with a thermodynamic model is needed because equilibrium partitioning of $HNO_3$ to $NO_3^-$ is RH and acidity dependent. Therefore, the high correlation of $NO_3^-$ with RH doesn't necessarily mean $NO_3^-$ was produced aqueously by uptake of $N_2O_5$. If the authors mean uptake of $HNO_3$ instead (L20) then the process is not aqueous chemistry, but rather shifts in equilibrium partitioning and dissolution of $HNO_3$.

We basically agree with the reviewer's points. However, the term "aqueous chemistry" we used here was actually organonitrate formation by non-radical reactions of glyoxal and $HNO_3$ in aerosol liquid water (Lim et al., 2016). Lim et al. (2016) showed that photochemical formation of $HNO_3$ through reactions of peroxy radicals, $NO_x$, and OH in gas-phase and $HNO_3$ uptake into the aerosol liquid water can enhance the organonitrate formation via reactions of glyoxal and $HNO_3$ in aqueous-phase. Organonitrates can also give $HNO_3$ by hydrolysis, and aerosol liquid water helps aerosol partitioning of $HNO_3$. Recent study by Lim et al. (manuscript in preparation, 2017) shows the hygroscopic growth and acidification of $PM_{2.5}$ in Seoul through the $HNO_3$ uptake–organonitrate formation–water uptake cycles, by using the combined kinetic and thermodynamic model simulation with humid chamber experiments and measurement data.

Instead of removing L17–22 on p.11, therefore, we modified the sentences as follows:

The high correlations of RH with nitrate ($r = 0.53$) (Table 5) implies a role of aqueous chemistry in nitrate aerosol formation in Seoul. A recent thermodynamic model simulation with smog chamber experiments and measurement data (Lim et al., in preparation, 2017) reveals that $NO_x$ photochemistry under the high RH condition facilitates hygroscopic growth of aerosols through water and $HNO_3$ uptake cycles with organonitrate formation in aerosol liquid water (Lim et al., 2016) and thus contributes to the high nitrate proportion to $PM_{2.5}$ in Seoul (Fig. 4a).

16. p. 11, L26: it was indicated on L15 (P11) that $NO_2$ decreased during 2/27-3/2 with high wind speeds, but that's not similar to the observations in CO that showed little difference with wind speed. Please clarify.

CO in Seoul showed smaller difference with wind speed compared to $NO_2$ in Seoul because it is also affected by regional transport as shown as CO in Deokjeok. Therefore, the difference of CO between Seoul and Deokjeok can be more like $NO_2$ in Seoul rather than CO in Seoul itself. To clarify this, L25–27 on p.11 was modified as follows:

On the other hand, the concentration differences of CO between Seoul and Deokjeok are large during the haze period but small on the high wind speed days (e.g. February 27 and March 2) like $NO_2$ concentrations, and this indicates local influence on the high CO levels in Seoul.

17. p. 11, L31-33: It's unclear why trends in organic tracers suggests regional transport of $SO_2$.

L31–33 on p.11 are now modified as follows:

These are the local characteristics as shown in $NO_2$. Higher concentrations of *n*-alkanes, monocarboxylic acids, and sugars during the haze period than those during the clean period in Deokjeok show the regional transport characteristics as shown in $SO_2$.

18. p. 12, L7: Why is the O:C of carboxylic acids only explored? The conclusions drawn are acceptable if O:C represented values for all components of OA.

We newly calculated O/C for the total identified OM (using analyzed mono- and dicarboxylic acids, sugars, *n*-alkanes, and PAHs all together, of which mass is ~5% of total estimated OM, Fig. 7a) and represented its daily time series in Fig. 7h. Day-to-day variation of the O/C of total identified OM is mostly contributed by that of dicarboxylic acids O/C. On the other hand, small variability of O/C in monocarboxylic acids and sugars (Fig. S2) implies that the sources of these two organic compound groups are mainly primary rather than secondary production.

We modified L7–8 on p.12 as follows:

The O/C of dicarboxylic acids and total identified OM that stayed high during both early and late stages of the haze period in both two places indicate secondary production as a source of the organic compounds during the long-lasting haze (Figs. 7g and 7h).

[Figure]

**Figure S2: Daily time-series of O/C of (a) sugars, (b) monocarboxylic acids, and (c) dicarboxylic acids at KIST site in Seoul and Deokjeok site for the analysis period.**

19. p. 12, L12-13: I disagree with the conclusion here. If RH and OC are not well correlated while RH and OC/EC is, this suggests to me that EC and RH are somehow correlated, but still EC is solely a primary tracer, I think the correlation is merely due to meteorology and cannot suggest anything about aq-phase production of OC.

Total OC can be regarded as the sum of primary OC (POC) and secondary OC (SOC). POC can be represented with a constant intrinsic OC-to-EC ratio of local primary emission ($[OC/EC]_{pri}$) (e.g. Castro et al., 1999) as follows:
$OC = POC + SOC = [OC/EC]_{pri} \times EC + SOC$
Thus, the variability of OC/EC represents that of SOC/EC, and correlations of them with RH should be same ($r = +0.51$ in this study). This value is much higher than correlations of EC (and POC, $r =$

+0.03), OC ($r$ = +0.33), and SOC (approximate SOC calculated using the minimum OC/EC of 2.9 as [OC/EC]$_{pri,}$ see the modified Table 1, $r$ = +0.39) with RH.

Weak correlation of EC with RH is natural because EC is a primary tracer, and anthropogenic emissions do not depend on the meteorological condition. The correlation of SOC with RH could also be lower than that of SOC/EC (or OC/EC) because the amount of SOC not only depends on chemistry but also depends on the amount of precursors. Since the precursor emissions could be approximately inferred from the primary pollutant marker, EC, the SOC (or OC) weighted by EC (SOC/EC or OC/EC) shows the degree of secondary production of OC. Therefore, the high correlation between OC/EC and RH shows probable influence of RH on the degree of SOC production. Since O/C (and OM/OC) increase by photochemical processing and secondary organic aerosols (SOA) production (Aiken et al., 2008), and hygroscopicity increases with increasing O/C (Jimenez et al., 2009), the higher correlations of RH with O/C ($r$ = +0.54) and OM/OC ($r$ = +0.51) compared to that of other meteorological variables (see the modified Table 5) support that aqueous process played a role in the SOA production during the measurement period.

Therefore, instead of removing L12–13 on p.12, we added a sentence at the end of the paragraph as follows:

Hygroscopicity of OA can be enhanced with increasing O/C and OM/OC by SOA production and aging (Aiken et al., 2008; Jimenez et al., 2009). The higher correlations of RH with O/C of total identified OM ($r$ = 0.54) and OM/OC ($r$ = 0.51) compared to that of other meteorological variables implies hygroscopic properties of SOA produced during the long-lasting haze in Seoul.

**Figures**

Fig. 2- please add the sampling site locations to one of the panels.

We marked locations of Seoul and Deokjeok sampling sites in the panels, as the reviewer suggested.

Fig. 3- I understand that OC was the parameter directly measured, but since the pie charts represent total aerosol mass from each species, why not convert OC to OM, using appropriate, representative ratios of OM/OC?

We modified the pie charts as following the referee's suggestion. Average OM concentrations in the pie charts (Fig. 4) were derived from daily measured OC and calculated OM/OC ratios of identified OM (~5% of estimated OM).

Fig. 6- I suggest marking the political boundaries with a different colors than the contours. Also, the wind arrows don't show up well. Consider making the panels larger. It would also help if the locations of the sampling sites are marked on one panel.

We enlarged each panel in Fig. 8 and modified figures following the reviewer's suggestion.

**References**

Aiken, A. C., Decarlo, P. F., Kroll, J. H., Worsnop, D. R., Huffman, J. A., Docherty, K. S., Ulbrich, I. M., Mohr, C., Kimmel, J. R., Sueper, D., Sun, Y., Zhang, Q., Trimborn, A., Northway, M., Ziemann, P. J., Canagaratna, M. R., Onasch, T. B., Alfarra, M. R., Prevot, A. S. H., Dommen, J., Duplissy, J., Metzger, A., Baltensperger, U., and Jimenez, J. L.: O/C and OM/OC ratios of primary, secondary, and ambient organic aerosols with high-resolution time-of-flight aerosol mass spectrometry, Environ. Sci. Technol., 42, 4478–4485, 2008.

Castro, L. M., Pio, C. A., Harrison, R. M., and Smith, D. J. T.: Carbonaceous aerosol in urban and rural European atmospheres: estimation of secondary organic carbon concentrations, Atmos. Environ., 33, 2771–2781, 1999.

Choi, N. R., Lee, S. P., Lee, J. Y., Jung, C. H., and Kim, Y. P.: Speciation and source identification of organic compounds in $PM_{10}$ over Seoul, South Korea, Chemosphere, 144, 1589–1596, 2016.

Falkovich, A. H., Graber, E. R., Schkolnik, G., Rudich, Y., Maenhaut, W., and Artaxo, P.: Low molecular weight organic acids in aerosol particles from Rondônia, Brazil, during the biomass-burning, transition and wet periods, Atmos. Chem. Phys., 5, 781–797, 2005.

Gentner, D. R., Jathar, S. H., Gordon, T. D., Bahreini R., Day, D. A., El Haddad, I., Hayes, P. L., Pieber, S. M., Platt, S. M., de Gouw, J., Goldstein, A. H., Harley, R. A., Jimenez, J. L., Prévôt, A. S. H., and Robinson, A. L.: Review of urban secondary organic aerosol formation from gasoline and diesel motor vehicle emissions, Environ. Sci. Technol., 51, 1074–1093, 2017.

Grosjean, D.: Organic Acids in Southern California Air: Ambient Concentrations, Mobile Source Emissions, in Situ Formation and Removal Processes, Environ. Sci. Technol., 23, 1506–1514, 1989.

Jimenez, J. L., Canagaratna, M. R., Donahue, N. M., Prevot, A. S. H., Zhang, Q., Kroll, J. H., DeCarlo, P. F., Allan, J. D., Coe, H., Ng, N. L., Aiken, A. C., Docherty, K. S., Ulbrich, I. M., Grieshop, A. P., Robinson, A. L., Duplissy, J., Smith, J. D., Wilson, K. R., Lanz, V. A., Hueglin, C., Sun, Y. L., Tian, J., Laaksonen, A., Raatikainen, T., Rautiainen, J., Vaattovaara, P., Ehn, M., Kulmala, M., Tomlinson, J. M., Collins, D. R., Cubison, M. J., Dunlea, E. J., Huffman, J. A., Onasch, T. B., Alfarra, M. R., Williams, P. I., Bower, K., Kondo, Y., Schneider, J., Drewnick, F., Borrmann, S., Weimer, S., Demerjian, K., Salcedo, D., Cottrell, L., Griffin, R., Takami, A., Miyoshi, T., Hatakeyama, S., Shimono, A., Sun, J. Y., Zhang, Y. M., Dzepina, K., Kimmel, J. R., Sueper, D., Jayne, J. T., Herndon, S. C., Trimborn, A. M., Williams, L. R., Wood, E. C., Middlebrook, A. M., Kolb, C. E., Baltensperger, U., and Worsnop, D. R.: Evolution of Organic Aerosols in the Atmosphere, Science, 326, 1525–1529, 2009.

Kawamura, K. and Kaplan, I. R.: Motor exhaust emission as a primary source of dicarboxylic acids in Los Angeles ambient air, Environ. Sci. Technol., 21, 105–110, 1987.

Kundu, S., Kawamura, K., Andreae, T. W., Hoffer, A., and Andreae, M. O.: Molecular distributions of dicarboxylic acids, ketocarboxylic acids and α-dicarbonyls in biomass burning aerosols: implications for photochemical production and degradation in smoke layers, Atmos. Chem. Phys., 10, 2209–2225, 2010.

Lee, J., Cho, K. S., Jeon, Y., Kim, J. B., Lim, Y.-R., Lee, K., and Lee, I.-S.: Characteristics and distribution of terpenes in South Korean forests, J. Ecol. Environ., 41, 1–19, 2017.

Lim, Y. B., Kim, H., Kim, J. Y., and Turpin, B. J.: Photochemical organonitrate formation in wet aerosols. Atmos. Chem. Phys., 16, 12631-12647, 2016.

Narukawa, M., Kawamura, K., Takeuchi, N., and Nakajima, T.: Distribution of dicarboxylic acids and carbon isotopic compositions in aerosols from 1997 Indonesian forest fires, Geophys. Res. Lett., 26(20), 3101–3104, 1999.

Oros, D. R. and Simoneit, B. R. T.: Identification and emission factors of molecular tracers in organic aerosols from biomass burning, part 2, Deciduous trees, Appl. Geochem., 16, 1545–1565, 2001.

Rogge, W. F., Hildemann, L. M., Mazurek, M. A., Cass, G. R., and Simoneit, B. R. T.: Sources of fine organic aerosol. 1. Charbroilers and meat cooking operations, Environ. Sci. Technol., 25, 1112–1125, 1991.

Rogge, W. F., Hildemann, L. M., Mazurek, M. A., Cass, G. R., and Simoneit, B. R. T.: Sources of fine organic aerosol. 9. Pine, oak, and synthetic log combustion in residential fireplaces, Environ. Sci. Technol., 32, 13–22, 1998.

Stogiannidis, E. and Laane, R.: Source characterization of polycyclic aromatic hydrocarbons by using their molecular indices: An overview of possibilities, in: Reviews of Environmental Contamination and Toxicology, Springer International Publishing, Switzerland, 49–133, 2015.

---

## Author Comment (AC2) · 20 Jun 2017

**Response to Anonymous Referee #2**

This paper discusses the results of a study of $PM_{2.5}$ bound inorganic - organic species and gaseous pollutants on the haze events in February 2014 at the outflow regions of Asian pollution. Meteorological factors and synoptic conditions are also discussed along with chemistry for the same context. These data of Asian outflow were collected at a site in highly industrialized region of Seoul and at a background site, Deokjeok Island, over the Yellow Sea. Paper contents important data and interesting discussions on the topic, hence I recommend for the publication in ACP. However, I have several comments/suggestions, especially at several places where statements are contradictory which should be addressed before making a final decision.

We thank the reviewer for the valuable comments and constructive suggestions. Following the reviewer's comments and suggestions, texts, figures, and tables in the original manuscript have been modified. In addition, we have conducted additional analyses, modified texts, figures, and tables, and added new figures (Figs. 2 and 5) and references in the revised manuscript. Each response to reviewer colored in blue and changes in the manuscript colored in red.

P1, L16-18: Not clear. It is an overall general statement for both the sites. Better to summarise according to individual site, as in the following sentences.

The sentence is an overall description for the haze in Seoul. To clarify it, we inserted "in Seoul" into the sentence.

Dominance of fine-mode particles ($PM_{2.5}/PM_{10} \sim 0.8$), a large secondary inorganic fraction (76%), high OC/EC ($> 7$), and highly oxidized aerosols (oxygen-to-carbon ratio of $\sim 0.6$ and organic mass-to-carbon ratio of $\sim 1.9$) under relatively warm, humid, and stagnant conditions characterize the multi-day haze episode in Seoul

P4, L3-10: Do authors think that it is logically correct to measure $PM_{2.5}$ mass using low- volume sampler and compare $PM_{2.5}$ chemical measurement using high-volume sampler. Possible biases should be properly addressed here with references (e.g. MAPAN, 2013, Volume 28, Issue 3, pp 153–166).

We agree with the reviewer's comments. As summarized in Aggarwal et al. (2013), Lagler et al. (2011) reported underestimation of PM concentrations measured by high-volume air sampler. In addition, there also could be gas-phase artifacts in the OC measurement in this study, as the other reviewer pointed out. Considering both (i) difference between the high- and low- volume samplers and (ii) possible positive and negative artifacts of OC concentration, we added a new paragraph at the end of Sect. 2.1 as follows:

Note that the $PM_{2.5}$ sampling conducted in this study could result in artifacts of carbonaceous species. Firstly, we used the high-volume air sampler for carbon analyses, while the total mass and ion concentrations were obtained by the low-volume air sampler. Secondly, we did not employ preceding organic denuder and backup filters in the high-volume air sampler for correction of both positive artifacts (by adsorption of organic vapor) and negative artifacts (by volatilization of semi-volatile materials) of measured OC. Although the positive artifacts are thought to be larger than the negative artifacts (Chow et al., 2010; Kim et al., 2016), the high-volume air sampler tends to underestimate particle concentrations (Lagler et al., 2011; Aggarwal et al., 2013) and thus the measurement by the high-volume air sampler may partly reduce such positive artifacts as recently estimated by Kim et al. (2016). However, aging and oxidation properties and source characteristics of measured aerosols can still be altered by potential loss of the semi-volatile organic compounds. In this study, therefore, we used organic compounds more for qualitative comparisons between different places and periods rather than for quantitative analysis.

P4, L12-13: The readability/ sensitivity of microbalance should be mentioned here.

The precision of the microbalance we used was 1 µg. We added this information within the sentence as follows:

The mass concentrations of $PM_{2.5}$ and $PM_{10}$ were measured by the Mettler MT5 microbalance (Mettler-Toledo, Greifensee, Switzerland) with a precision of 1 µg after 24 h standing of the Teflon filter sample in a desiccator.

P6, L13-16: Back trajectory analysis should also be discussed in support.

We added new figures of backward trajectories for the haze and clean periods as Fig. 2 and modified L13–17 in p.6 as follows:

The wind direction during the overall analysis period was mostly westerly or west-northwesterly, except easterly winds on February 26. Considering negligible local emissions in and near the Deokjeok Island, the high $PM_{2.5}$ level in Deokjeok is most likely due to the regional transport from China, evidenced by the high $PM_{2.5}$ levels in the upwind Chinese cities (Fig. 1c) and backward trajectories during the haze period (Fig. 2a). On the other hand, the high $PM_{2.5}$ level in Deokjeok on February 26 seems to result from the easterly transport of pollutants from the SMA, as shown by backward trajectories passing through the SMA before reaching Deokjeok (Fig. 2b).

[Figure]

**Figure 2: Backward trajectories from 500 m above sea level over the KIST site in Seoul (red) and the Deokjeok site (blue) at 09:00 and 21:00 local time (GMT+0900) during (a) the early stage of haze, (b) the late stage of haze, and (c) clean period.**

P6, L17-18: Not clear.

To make the sentence clear, L17–18 in p.6 was modified as follows:

The high $PM_{2.5}$ levels in Seoul but the low $PM_{2.5}$ concentrations in Deokjeok on the following 3 days (Fig. 1b) as well as the stagnant backward trajectories (Fig. 2b) show that the prolonged haze period in Seoul did not only result from the transboundary transport of pollutant but could be also affected by local sources.

P6, L33: "compared and characterized in Table 4" should be "compared and summarised in Table 4"

Thanks for correction. We replaced "characterized" with "summarized."

P7, L3: Please mention (number) boundary layer height and wind speed here.

We inserted approximate values of boundary layer height (~400 m) and wind speed (~2 m s$^{-1}$) for the haze period into the sentence as the review suggested.

P7, L7-10: Backward trajectory should also be discussed here.

With the new figure set (Fig. 2), we modified the sentence as follows:

Therefore, the regional influences on the severe haze can be inferred from the high concentration of each chemical component (Table 1) as well as the backward trajectories from China (Figs. 2a–b) during the haze period in both Seoul and Deokjeok.

P7, L12-18: Section 3.2.2, this is in contrast to section 3.2.1 and following sections, e.g. P13, L10-12, and several other places (please see comments below).

In Sect. 3.2.1, we wrote that "regional influence" can be inferred from high concentrations of each chemical component in both Seoul and background and "local influence" can be inferred from the higher concentration of each chemical component in Seoul than those in background. In Sect. 3.2.2, we insisted that $SO_2$ in Seoul is affected by "regional transport" since $SO_2$ was high in both sites during the haze. On the other hand, $NO_2$ was significantly higher in Seoul than Deokjeok, and thus we insisted that $NO_2$ in Seoul is affected by "local emissions." In the sentences L10–13 on p.13, we insisted again that $NO_2$ and nitrate were higher in Seoul than Deokjeok and these species are related to "local emissions and production." We didn't find contrasting points among these three parts.

P8, L1-2: How about the contribution of marine aerosols, especially at Deokjeok? Mass concentration of $PM_{2.5}$ and $PM_{10}$ should also consider while discussing $PM_{2.5}/PM_{10}$ ratios.

As shown by extremely low concentrations of $Na^+$ (~0.1 µg m$^{-3}$) in Deokjeok and Seoul during both haze and clean period (Table 1), marine aerosols seem to contribute little to observed $PM_{2.5}$ in this study. Sea salt estimated by known mass ratio of ions to $Na^+$ in seawater (0.25 for $SO_4^{2-}$, 0.037 for $K^+$, 0.038 for $Ca^{2+}$, and 0.12 for $Mg^{2+}$; Berg and Winchester, 1978) during the haze was ~0.4 µg m$^{-3}$ in Seoul and ~0.5 µg m$^{-3}$ in Deokjeok (Table S1). These values are negligible compared to the $PM_{2.5}$ mass concentrations of 116 µg m$^{-3}$ in Seoul and 84 µg m$^{-3}$ in Deokjeok.

Brief discussion about $PM_{10}$ and $PM_{2.5}$ mass concentrations during the measurement period can be found in Sect. 3.1. $PM_{10}$ was not much considered in this study because we focused on the fine mode particles, which occupied more than 80% of $PM_{10}$ during the haze and thus induced the multi-day haze event.

**Table S1: The average and standard deviation of PM$_{2.5}$ non-sea salt (nss-) ions and sea salt in Seoul and Deokjeok for the haze (February 23–28, 2014) and clean (March 5–9, 2014) periods.**

| Components | Seoul | | Deokjeok | |
|---|---|---|---|---|
| | Haze | Clean | Haze | Clean |
| nss-SO$_4^{2-}$ (µg m$^{-3}$) | 34.9 ± 9.1 | 3.9 ± 1.4 | 29.2 ± 12.3 | 4.7 ± 2.6 |
| NO$_3^-$ (µg m$^{-3}$) | 32.8 ± 8.4 | 4.6 ± 4.2 | 11.4 ± 8.5 | 2.8 ± 3.0 |
| nss-Cl$^-$ (µg m$^{-3}$) | 0.9 ± 0.4 | 0.2 ± 0.2 | 0.4 ± 0.5 | −0.1 ± 0.1 |
| NH$_4^+$ (µg m$^{-3}$) | 21.6 ± 4.3 | 2.7 ± 1.6 | 14.4 ± 6.0 | 2.4 ± 1.8 |
| nss-K$^+$ (µg m$^{-3}$) | 0.9 ± 0.2 | 0.2 ± 0.1 | 0.7 ± 0.3 | 0.2 ± 0.1 |
| nss-Ca$^{2+}$ (µg m$^{-3}$) | 0.3 ± 0.1 | 0.1 ± 0.0 | 0.2 ± 0.1 | 0.1 ± 0.0 |
| nss-Mg$^{2+}$ (µg m$^{-3}$) | 0.1 ± 0.0 | 0.1 ± 0.0 | 0.1 ± 0.0 | 0.1 ± 0.1 |
| Sea salt (µg m$^{-3}$) | 0.4 ± 0.1 | 0.2 ± 0.1 | 0.5 ± 0.2 | 0.5 ± 0.3 |

P8, L15: For more clarity, it is better to also use nss-SO$_4^{2-}$ concentration in discussions.

Non-sea salt (nss-) ionic components were estimated by a simple calculation, which assumes that the sea salt contributes Na$^+$ in PM$_{2.5}$ alone (George et al., 2008), as follows:
$M_{nss-X} = M_X − (X/Na^+)_{SW} × M_{Na}$
where, $M_X$ is the total mass of a species X, $(X/Na^+)_{SW}$ is a mass ratio of X to Na$^+$ in seawater, $M_{Na}$ is the mass of Na$^+$, and $M_{nss-X}$ is the mass of the nss- component in $M_X$. The ratio of $(X/Na^+)_{SW}$ for each ion was mentioned above, and the nss-ion concnetrations were summarized in Table S1. Since the concentrations of nss-SO$_4^{2-}$ are nearly identical to SO$_4^{2-}$ as shown in Table S1, we used SO$_4^{2-}$ rather than nss-SO$_4^{2-}$ in this study.

P8, L20-21: OC/EC ratios in Deokjeok and Seoul are 7.4±1.7 and 7.3±1.1, respectively in haze period. Do authors think "The OC/EC ratio is higher in Deokjeok than in Seoul during both haze and clean periods." is a correct sentence if authors see the values in view of statistical significance? Similarly check such statements in other places as well.

We agreed with the reviewer's comment. In the revised version of manuscript, we estimated approximate primary OC (POC) and secondary OC (SOC) using simple EC tracer method (Castro et al., 1999) and added them on Table 1. For the analysis period, the minimum OC/EC ratios in Seoul and Deokjeok are respectively 2.88 and 2.18. We assumed that these values as the primary OC/EC ratios ([OC/EC]$_{pri}$) and calculated POC (= [OC/EC]$_{pri}$ × EC) and SOC (= OC – POC). Since the more secondary fraction in OC in Deokjeok compared to that in Seoul can be represented better by SOC/OC, we replaced OC/EC with SOC/OC in the text.

With the brief description for estimation of POC and SOC, L18–22 on p.8 is now modified as follows:

Approximate primary OC (POC) and secondary OC (SOC) obtained by simple EC tracer method (Castro et al., 1999) using the minimum OC/EC values of 2.9 for Seoul and 2.2 for Deokjeok on March 9 as the primary OC/EC. As shown in Table 1, EC and thus POC concentrations in Seoul during both haze and clean periods are higher than those in Deokjeok, like CO. This shows that EC and POC in Seoul is largely contributed by local emissions while that in Deokjeok is mostly influenced by regional transport. Although OC concentration is higher in Seoul than that in Deokjeok, its secondary fraction (SOC/OC) is higher in Deokjeok than in Seoul during both haze and clean periods probably due to the secondary production during the long-range transport to the background, as discussed in Sect. 3.3.

P9, L4-16: Discussion is contradictory in context of sources discussed before this para and later in the text.

Since various diagnostic ratios of ambient organic compounds represent mixture of various source emissions, it is hard to avoid some contradiction and ambiguousness in their interpretation. In addition, there could be loss of semivolatile low molecular weight compounds during the measurement, strict interpretation of sources with reported diagnostic ratio could be inaccurate. Therefore, we here tried to compare the PAH ratios of different sites and different periods and to interpret them relatively. Following another reviewer's suggestion, we introduce a new figure (Fig. 5) and modified this paragraph (from the third sentence in the paragraph) as follows:

The average PAH ratios in Seoul and Deokjeok for the haze and clean periods are summarized in Fig. 5 with previously reported diagnostic ratios (Yunker et al., 2002; Pies et al., 2008; De La Torre-Roche et al., 2009; Akyüz and Çabuk, 2010; Oliveira et al., 2011). Interpretation of these PAH ratios all together seems ambiguous and contradictory due to mixing of various emission sources. However, relative comparisons of each ratio between different places and different periods reveal more pyrogenic sources such as fossil fuel combustion and vehicular emissions in Seoul than in Deokjeok (Figs. 5a–c), in the overall influence of coal combustion and/or biomass burning during the haze period (Figs. 5a–b, and 5d).

As wrote in this paragraph, more pyrogenic sources (fossil fuel combustion and vehicular emissions) in Seoul and overall influence of coal combustion and biomass burning during the haze were identified. As described in the next paragraph, levoglucosan shows higher concentration in both site during the haze, and this also indicate biomass burning during the haze.

[Figure]

**Figure 5: Various PAH ratios of (a) FLA/(FLA + PYR), (b) BaA/(BaA + CHR), (c) ANT/(ANT + PHE), (d) IcdP/(IcdP + BghiP), and (e) BaP/(BaP + BeP) in Seoul (red diamonds) and Deokjeok (blud squares) during the haze (filled symbols) and clean (opened symbols) periods. Horizontal bars indicate standard deviation.**

P9, L18: "total sugar" should be "total sugar identified"

Thanks, we corrected it as following the reviewer's suggestion.

P9, L31: Reference is needed.

The original sentence was modified and moved into the paragraph with references, as the third and fourth sentences of the paragraph as follows:

Whereas short-chain (light) *n*-alkanes are mostly associated with incomplete combustion or vehicle exhaust (Gentner et al., 2017), biosynthetic processes result in long-chain high molecular weight ($C_{27}$–$C_{33}$) *n*-alkanes with distinctive odd-to-even carbon number preference (Simoneit, 1991; Rogge et al., 1993). Note that *n*-alkanes in this study reflect less anthropogenic sources because we analyzed only high molecular weight *n*-alkanes ($C_{20}$–$C_{36}$).

P9, L33; P10, L1-2: Again contradictory statements (as pointed out above).

The sentence from L33 on p.9 to L2 on p.10 is modified as follows:

However, Seoul is more affected by fossil fuel combustion and vehicular emissions compared to Deokjeok and seems to also have local biomass burning and biogenic emission sources.

P10, L10: "OC/EC ratio" same comment as in P8, L20-21.

We replaced "OC/EC ratio" with "SOC/OC" in the revised version.

P10, Section 3.3.2: Result suggests that particles are fresh in Seoul and comparatively aged in Deokjeok. An analysis of fresh and aged $PM_{2.5}$, $PM_{10}$ should be incorporated.

In general, aging and oxidation properties of OA is widely analyzed by its atomic oxygen-to-carbon ratios (O/C) and organic mass (OM)-to-OC ratios (OM/OC) (e.g.; Aiken et al., 2008; Jimenez et al., 2009). In the revised version, we newly estimated total OM from the identified OM (*n*-alkanes, PAHs, mono- and dicarboxylic acids, and sugars) and obtained OM/OC as well as O/C. We added average concentrations of OM with average values of OM/OC and O/C in Table 2 and inserted following sentences into the first paragraph of Sect. 3.3:

To explore the oxidation and aging properties of OA, atomic oxygen-to-carbon ratios (O/C) and organic mass-to-carbon ratios (OM/OC) of SOA have been widely used (Aiken et al., 2008; Jimenez et al., 2009). The organic mass (OM) identified in current study is ~5% of total OM estimated by OM/OC of the identified OM.

Also, we added following paragraph, discussing aging properties of observed particles in Seoul with OM/OC and O/C of identified OM, into Sect. 3.3.2 as its first paragraph:

Average OM/OC and O/C in Seoul were 1.92 and 0.59 for the haze period and 1.83 and 0.52 for the clean period (Table 2). Although these values may be overestimated because of possible loss of semivolatile compounds arising from the measurement and analysis addressed in Sects. 2.1 and 2.2, the higher values of OM/OC and O/C during the haze indicate OA was more oxygenated. Considering the higher OM/OC and OC in Deokjeok of 2.00 and 0.65 for the haze period (Table 2) together with the backward trajectories from China during the haze (Fig. 2), haze particles were already aged before arriving at Seoul and mixed with fresh compounds in Seoul.

Since we focused more on $PM_{2.5}$ rather than $PM_{10}$, various chemical analyses for $PM_{10}$ sample were not conducted, and thus additional analyses based on $PM_{10}$ data are not possible. Considering the high

PM$_{2.5}$/PM$_{10}$ ratios during the haze period (> 0.8), however, the aging properties of PM$_{10}$ in Seoul and Deokjeok are probably similar to that of PM$_{2.5}$.

P11, L5-8: How about boundary layer height?

Obviously the stable and stagnant environment during the haze (low winds and shallow boundary layer) could help the secondary formation of aerosols. We added correlations of SOR with wind speed and boundary layer height in L4–9 on p.11 as follows:

Along with the stable and stagnant environment, the warm and humid air conditions during the haze period (Fig. 6e) could be also conducive to both gas-phase and aqueous-phase oxidation of SO$_2$ (Liang and Jacobson, 1999; Seinfeld and Pandis, 2006). SOR shows significant correlations with temperature ($r = 0.72$) or RH ($r = 0.59$) as well as wind speed ($r = -0.64$) and BLH ($r = -0.88$).

P11, L10-22: I suggest to check and discuss the relation of RH, NO$_2$ with sulfate formation apart from photochemical formation.

Recent measurement study on the Chinese haze suggested that the aqueous SO$_2$ oxidation by NO$_2$ is a key process of sulfate formation in high RH and high SO$_2$, NO$_x$, and NH$_3$ condition, and this process promotes to nitrate and OM formation on aqueous particle (Wang et al., 2016). If we considered only Seoul, the aqueous SO$_2$ oxidation by NO$_2$ seems to be plausible because correlations of SO$_4^{2-}$ with RH ($r = 0.47$), SO$_2$ ($r = 0.87$), NO$_2$ ($r = 0.81$), and NO$_3^-$ ($r = 0.96$) in Seoul were high enough. However, in fact, the high SO$_4^{2-}$ in Seoul during the haze mostly came from the regional transport rather than the local formation, as shown by the high SO$_4^{2-}$ but extremely low NO$_2$ in Deokjeok during the same period (Fig. 6a and Table 1). If we made a crude assumption that SO$_4^{2-}$ concentration differences between Seoul and Deokjeok represents locally produced SO$_4^{2-}$, much lower correlations of it with SO$_2$ ($r = 0.35$), NO$_2$ ($r = 0.42$), and NO$_3^-$ ($r = 0.58$) in Seoul, and comparable correlation of it with RH ($r = 0.51$) in Seoul do not quite support a role of such aqueous SO$_2$ oxidation by NO$_2$ in Seoul.

P12, L1-2: I suggest to check and discuss primary and secondary OC contribution in haze and clean days at both the site to ustify this statement.

Although the POC and SOC values estimated by the EC tracer method are very crude, the local influence of POC on the high OM concentrations in Seoul during the haze can be inferred from the higher POC concentration with high POC/OC proportion in Seoul (5.7 µg m$^{-3}$, 40% of OC) than those in Deokjeok (2.5 µg m$^{-3}$, 31% of OC). Note that proportions of POC in OC in Seoul and Deokjeok are much higher during the clean period (81% in Seoul and 53% in Deokjeok) without regional transport of primary pollutants from China.

Thus, we modified L1–2 on p.12 as follows:

And this clearly indicate local influence of primary organic compounds on the high OM concentrations in Seoul during the prolonged haze, as also shown by the higher proportion of POC in OC as well as the higher concentration of POC in Seoul (5.7 µg m$^{-3}$, 40% of OC,), compared with those in Deokjeok (2.5 µg m$^{-3}$, 31% of OC) (Table 1).

P13, L10-12: Not clear.

To make it clear, the sentence is modified as follows:

The high concentrations of SO$_2$, sulfate, nitrate, CO, and OM in both Seoul and Deokjeok on these two days (Figs. 6a, 6c–d, and 7a) indicate quick transport of PM$_{2.5}$ components from China.

**References**

Aggarwal, S. G., Kumar, S., Mandal, P., Sarangi, B., Singh, K., Pokhariyal, J., Mishra, S. K., Agarwal, S., Sinha, D., Singh, S., Sharma, C., and Gupta, P. K.: Traceability issue in $PM_{2.5}$ and $PM_{10}$ measurements, Mapan-J. Metrol. Soc. I., 28, 153–166, 2013.

Berg Jr, W. W. and Winchester, J. W.: Aerosol chemistry of marine atmosphere, Chem. Oceanogr., 7, 173–231, 1978.

Castro, L. M., Pio, C. A., Harrison, R. M., and Smith, D. J. T.: Carbonaceous aerosol in urban and rural European atmospheres: estimation of secondary organic carbon concentrations, Atmos. Environ., 33, 2771–2781, 1999.

George, S. K., Nair, P. R., Parameswaran, K., Jacob, S., and Abraham, A.: Seasonal trends in chemical composition of aerosols at a tropical coastal site of India, J. Geophys. Res., 113, D16209, 2008.

Lagler, F., Belis, C., and Borowiak, A.: A quality assurance and control program for $PM_{2.5}$ and $PM_{10}$ measurements in European Air Quality Monitoring Networks (EUR 24851 EN), Joint Research Centre, Institute for Environment and Sustainability, Publications Office of the European Union, Luxembourg, 118 pp., 2011.

Wang, G., Zhang, R., Gomez, M. E., Yang, L., Zamora, M. L., Hu, M., Lin, Y., Peng, J., Guo, S., Meng, J., Li, J., Cheng, C., Hu, T., Ren, Y., Wang, Y., Gao, J., Cao, J., An, Z., Zhou, W., Li, G., Wang, J., Tian, P., Marrero-Ortiz, W., Secrest, J., Du, Z., Zheng, J., Shang, D., Zeng, L., Shao, M., Wang, W., Huang, Y., Wang, Y., Zhu, Y., Li, Y., Hu, J., Pan, B., Cai, L., Cheng, Y., Ji, Y., Zhang, F., Rosenfeld, D., Liss, P. S., Duce, R. A., Kolb, C. E., Molina, M. J.: Persistent sulfate formation from London Fog to Chinese Haze, P. Natl. Acad. Sci. USA, 113, 13630–13635, 2016.